# The Flemmingsome reveals an ESCRT-to-membrane coupling via ALIX/syntenin/syndecan-4 required for completion of cytokinesis

Cyril Addi[1,2], Adrien Presle[1,2], Stéphane Frémont[1], Frédérique Cuvelier[1], Murielle Rocancourt[1], Florine Milin[1], Sandrine Schmutz[3], Julia Chamot-Rooke [4], Thibaut Douché [5], Magalie Duchateau[5], Quentin Giai Gianetto[5,6], Audrey Salles[7], Hervé Ménager [6], Mariette Matondo [5], Pascale Zimmermann[8,9], Neetu Gupta-Rossi [1,10] & Arnaud Echard [1,10✉]

Cytokinesis requires the constriction of ESCRT-III filaments on the side of the midbody, where abscission occurs. After ESCRT recruitment at the midbody, it is not known how the ESCRT-III machinery localizes to the abscission site. To reveal actors involved in abscission, we obtained the proteome of intact, post-abscission midbodies (Flemmingsome) and identified 489 proteins enriched in this organelle. Among these proteins, we further characterized a plasma membrane-to-ESCRT module composed of the transmembrane proteoglycan syndecan-4, ALIX and syntenin, a protein that bridges ESCRT-III/ALIX to syndecans. The three proteins are highly recruited first at the midbody then at the abscission site, and their depletion delays abscission. Mechanistically, direct interactions between ALIX, syntenin and syndecan-4 are essential for proper enrichment of the ESCRT-III machinery at the abscission site, but not at the midbody. We propose that the ESCRT-III machinery must be physically coupled to a membrane protein at the cytokinetic abscission site for efficient scission, uncovering common requirements in cytokinesis, exosome formation and HIV budding.

[1] Membrane Traffic and Cell Division Lab, Institut Pasteur, UMR3691, CNRS, F-75015 Paris, France. [2] Sorbonne Université, Collège doctoral, F-75005 Paris, France. [3] Institut Pasteur, UTechS CB, F-75015 Paris, France. [4] Institut Pasteur, Mass Spectrometry for Biology Unit, C2RT, USR 2000, CNRS, F-75015 Paris, France. [5] Institut Pasteur, Proteomics Platform, Mass Spectrometry for Biology, C2RT, USR 2000, CNRS, F-75015 Paris, France. [6] Hub de Bioinformatique et Biostatistique – Département Biologie Computationnelle, Institut Pasteur, USR 3756 CNRS, F-75015 Paris, France. [7] UTechS Photonic BioImaging PBI (Imagopole), Centre de Recherche et de Ressources Technologiques C2RT, Institut Pasteur, Paris 75015, France. [8] Centre de Recherche en Cancérologie de Marseille (CRCM), Equipe labellisée Ligue 2018, Aix-Marseille Université, Inserm, CNRS, Institut Paoli Calmettes, 13009 Marseille, France. [9] KU Leuven, Department of Human Genetics, University of Leuven, B-3000 Leuven, Belgium. [10] These authors jointly supervised: Neetu Gupta-Rossi, Arnaud Echard. ✉email: arnaud.echard@pasteur.fr

Cytokinesis leads to the physical separation of daughter cells and concludes cell division. Final abscission occurs close to the midbody (or Flemming body), a prominent structure that matures at the center of the intercellular bridge connecting the two daughter cells and first described by Walther Flemming in 1891[1–9]. The scission occurs not at the midbody itself, but at the abscission site located at distance on one side of the midbody[10–14]. The first scission is usually followed by a second cleavage on the other side of the midbody, leaving a free MidBody Remnant (MBR)[8,12–15]. Then, MBRs are either released or wander, tethered at the cell surface for several hours, before being engulfed and degraded by lysosomes[14,16–19].

The endosomal sorting complexes required for transport (ESCRT) machinery plays a critical and evolutionarily conserved role in cytokinetic abscission, both in Eukaryotes and in Archea[20–31]. This machinery is composed of several protein complexes (ESCRT-0 to III) and culminates with the polymerization of filaments made of ESCRT-III components that contract in the presence of the ATPase VPS4 and ATP[31–35]. Remarkably, ESCRT-III-dependent helices of 17 nm filaments are observed at the abscission site by electron microscopy (EM), and ESCRT-III helical structures are often visible extending from the midbody to the abscission site[10,36]. Therefore, as in other topologically equivalent ESCRT-III-mediated events, including exosome biogenesis in multivesicular bodies (MVBs), retroviral budding or membrane repair, constriction of ESCRT-III filaments likely drives the final membrane scission during cytokinetic abscission[2–7,9].

The midbody plays a fundamental role in cytokinesis, as it constitutes a protein-rich platform that recruits key components for abscission, including the ESCRT machinery[2,3,6,9]. It is well established that the MKLP1 kinesin targets CEP55 to the midbody, which in turn recruits, through both ESCRT-I TSG101 and ESCRT-associated protein ALIX, the entire ESCRT machinery[29,37]. After this initial recruitment to the midbody itself and prior to abscission, the ESCRT-III machinery is progressively enriched to the future abscission site on the midbody side[12,13,20,21,25,30,31,36,38,39].

Since MKLP1 and CEP55 are only present at the midbody[38], it remains elusive how, mechanistically, ESCRT-III components can localize to the abscission site. Another crucial related issue is to reveal how the ESCRT-III filaments could be coupled to the plasma membrane, as final membrane constriction should require their tight association.

Here, we first set up an original method, using flow cytometry, for purifying intact post-cytokinetic MBRs, and identified 489 proteins enriched in this organelle by proteomics. Among them, we focused on the transmembrane protein syndecan-4 and associated proteins syntenin-1 (hereafter "syntenin") and ALIX, all highly enriched. Indeed, ALIX directly interacts with ESCRT-III[40] and we previously showed that syntenin can bind directly and simultaneously to the cytoplasmic tail of syndecans in vitro[41,42]. We thus hypothesized that ALIX-syntenin could mechanistically bridge the ESCRT machinery to the plasma membrane through the transmembrane proteoglycan syndecan-4. Interestingly, overexpression of syndecan-4 mutants that cannot be properly phosphorylated on the cytoplasmic tail was reported to perturb cytokinesis[43]. However, the underlying mechanism is unknown and whether syndecan-4 is actually required for cytokinesis has not been addressed. We here reveal that, together with ALIX, both syndecan-4 and syntenin are required for successful abscission in parallel to the TSG101 pathway, and promote the stable recruitment of the ESCRT-III machinery specifically at the abscission site.

## Results

**The Flemmingsome reveals candidates for abscission.** The proteome of intercellular bridges from CHO cells previously proved to be a particularly successful approach to identify proteins required for cytokinesis[44]. However, the use of detergents during the purification steps precluded the recovery of crucial proteins for cytokinesis, for instance the ESCRT components[44]. In order to purify intact midbodies without detergent treatment and thus reveal the complete proteome of these abscission platforms, we took advantage of the fact that released MBRs can be easily detached from the cell surface by EDTA treatment, as we previously reported[14]. Differential centrifugations helped to enrich intact MBRs from EDTA-treated HeLa cells ("Midbody Remnant Enriched fraction" or "MBRE", Fig. 1a) expressing the midbody-localized kinesin GFP-MKLP2[45] (which did not perturb the timing of abscission, Supplementary Fig. 1a). In order to improve the enrichment of MBRs, we developed an original protocol for isolating cell-free fluorescent GFP-positive MBRs ("MBR+") from EDTA-treated cells using flow cytometry (Fig. 1a, b; Supplementary Fig. 1b). In parallel, and as a control, we isolated small particles of matched size (1–3 μm) and granularity (SSC) but negative for GFP-MKLP2 ("MBR−") (Fig. 1a, b; Supplementary Fig. 1b). Western blot analysis demonstrated that the MBR+ population contained highly enriched known midbody proteins [MKLP1, CRIK, PRC1, PLK1, and CEP55] and showed reduced contamination, as compared to MBR−, total cell lysate (Tot) and MBRE fractions, with intracellular compartments [calreticulin (endoplamic reticulum), GM130 (golgi), Tom22 (mitochondria), HistoneH3 (nucleus), EEA1 (endosomes)] (Fig. 1c; Supplementary Fig. 1c). As expected, proteins such as ALIX, which participates both in cytokinesis and endosomal sorting in interphase were less enriched (twofold). Remarkably, cell membrane labeling with cell mask and scanning EM[46] further demonstrated that the MBR+ fraction contained membrane-sealed, intact MBRs (Fig. 1d), with very similar shape and length, as observed in vivo[14]. Immunofluorescence revealed that the 1–3 μm-sized objects sorted as MBR+ were indeed all GFP-positive MBRs and that the respective localization of key cytokinetic proteins [MKLP2, AuroraB, MKLP1, CEP55, RacGAP1, CHMP4B, ALIX, PRC1, and CRIK] was preserved (Fig. 1d). Thus, this original method to purify MBRs using flow cytometry sorting allowed us to obtain intact and highly pure MBRs, which correspond to post-abscission midbodies.

We next performed proteomic and statistical analysis to (1) identify proteins detected in seven independent MBR+ preparations and (2) identify proteins significantly enriched in these preparations, as compared to MBR−, MBRE, and/or total cell fractions. Since it is notoriously difficult to extract proteins from midbodies[47], we used sodium dodecyl sulfate (SDS) to fully solubilize proteins from our different fractions after purification. For mass spectrometry analysis, two methods for sample preparation were used and analyzed separately (SDS polyacrylamide gel electrophoresis (PAGE) gel/in-gel digestion and enhanced filter-aided sample preparation (eFASP[48])/in-solution digestion, and gave complementary results (Supplementary Fig. 2)). We detected a total of 1732 proteins with at least one unique identified peptide in the MBR+ preparations, constituting the *Total Flemmingsome* (Supplementary Data 1, TAB1), a name that we gave as a tribute to W. Flemming.

Among the 1732 proteins in MBR+, we defined as the *Enriched Flemmingsome* (Supplementary Data 1, TAB2) a subset of 489 proteins significantly enriched at least 1.3-fold with a false-discovery rate (FDR) < 5% as compared to MBRE, MBR−, or Tot (Fig. 1e, upper panel, Supplementary Fig. 1d and 2; Supplementary Data 1, TAB2-3) and/or quantitatively present in MBR+ but not detected in at least one other fraction (Fig. 1e, bottom panel; Supplementary

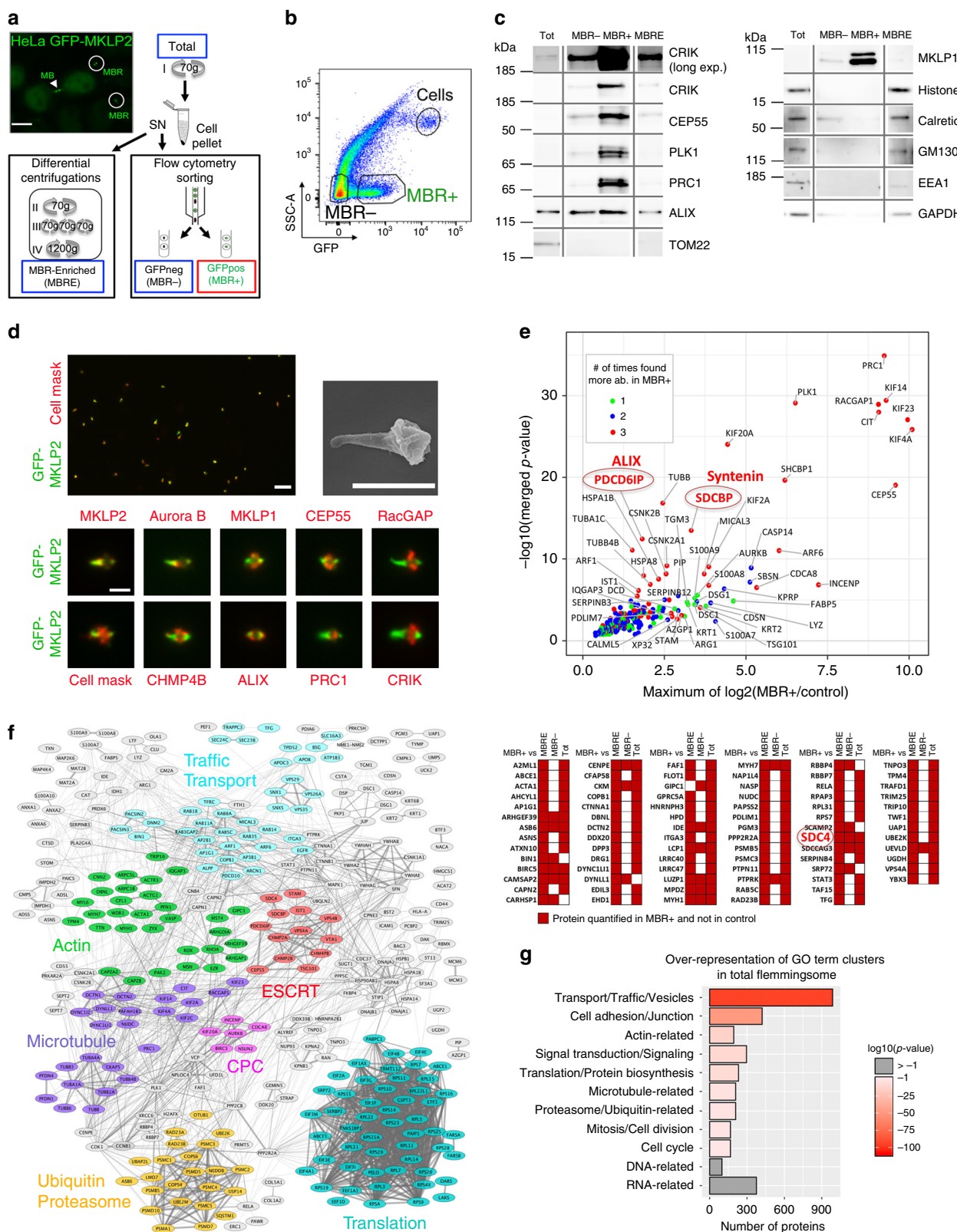

Data 1, TAB2, TAB4-5 and Methods). For instance, CRIK was found enriched >500-fold in MBR+ as compared to Total (Supplementary Data 1, TAB2, col I). Interestingly, differential analyses indicated that the most abundant and most significantly enriched proteins, such as MKLP1 (KIF23), MKLP2 (KIF20A), RacGAP1, KIF4A, PRC1, KIF14, PLK1, CEP55, and CRIK (CIT) corresponded to well established proteins of cytokinesis (Fig. 1e).

Volcano plots demonstrated that, independently of the extraction method (eFASP or gel-based), these core cytokinetic proteins were more enriched in MBR+ as compared to MBR−, MBRE, or Total (Supplementary Fig. 2), consistent with the results obtained by western blots (Supplementary Fig. 1c).

Interestingly, 150 out of the 489 proteins (31%) of the *Enriched Flemmingsome* have been already localized to the furrow, the

**Fig. 1 Proteomics of highly pure and intact post-abscission midbodies revealed known and previously unknown proteins enriched in this organelle.**
**a** Midbody remnant purification. HeLa cells (upper left picture) expressing GFP-MKLP2, a kinesin enriched in midbodies (MB) and midbody remnants (MBRs) were EDTA-treated (Total). After 70$g$ centrifugation, the supernatant (SN) containing MBRs was processed either (1) by differential centrifugations leading to MBR-enriched fraction (MBRE) or (2) subjected to flow cytometry sorting to purify GFP-positive MBRs (MBR+) and their GFP-negative counterpart (MBR−). **b** Representative pseudo-colored profile of flow cytometry sorting of MBRs. The MBR+ (14% total) and SSC-matched MBR− (44% total) were separated from remaining cells (1%). See Supplementary Fig. 1b. **c** Western blots of same amounts of protein extracts from Total (Tot), MBR-enriched (MBRE), flow cytometry-sorted MBR− and MBR+ populations. Membranes were blotted repeatedly with indicated antibodies. See also Supplementary Figs. 1c and 6. **d** Upper left panel: MBR+ population analyzed with cell mask membrane marker. Each individual midbody is positive for GFP-MKLP2 (green) and cell mask (red) Scale bar: 6 μm. Upper right panel: scanning electron microscopy of an isolated MBR. Note the intact and sealed membrane. Lower panels: immunofluorescence stainings of MBR+ for endogenous proteins or membrane marker (red), as indicated. Scale bars: 2 μm. **e** *The Enriched Flemmingsome*. Upper panel: merged volcano plot of the mass spectrometry analysis showing the maximum log2(fold change) in *x*-axis measured between MBR+ and the other fractions (MBRE, MBR−, or Total) and the corresponding −log10(merged p value) in *y*-axis. color code: proteins significantly enriched in MBR+ when compared with 3 (red), 2 (blue), or 1 (green) of the other fractions. Bottom panel: proteins quantitatively present in MBR+ but not detected in at least two of the other fractions. ALIX (PDCD6IP), syntenin (SDCBP) and syndecan-4 (SDC4) circled in red. **f** STRING functional association network for the *Enriched Flemmingsome*. See Supplementary Fig. 3 for details. **g** GO-term over-representation clusters in the *Total Flemmingsome*. The size of each bar (*x*-axis) corresponds to the number of proteins in each cluster and the red gradient the enrichment *p* values coming from hypergeometric tests. Gray: *p* value > 0.1.

bridge or the midbody and/or functionally involved in cytokinesis, according to our literature search (Supplementary Data 1, TAB2 and dedicated website https://flemmingsome.pasteur.cloud/). Proteins of the *Enriched Flemmingsome* were highly connected and based on the literature, many fell into known functional categories involved in cytokinesis, such as "actin-related", "chromosomal passenger complex-related", "microtubule-related", "traffic/transport-related", or "ESCRT-related" (Fig. 1f; Supplementary Fig. 3). GO-term analysis revealed that 87.5% of proteins identified in MBR+ fell into 11 GO term clusters (defined in Data 1, TAB7). The category "mitosis/cell division" was more significantly overrepresented in MBR+ as compared to either total identified proteins (Fig. 1g) or MBRE (Supplementary Fig. 1e). This indicates that flow cytometry-based MBR+ purification is superior to MBRE purification for obtaining proteins known to be implicated in cell division.

Thus, our approach was highly successful at identifying 150 known cytokinetic proteins and, importantly, revealed 339 additional candidates potentially involved in cytokinesis/abscission. In the rest of this study, we decided to focus on ALIX (PDCD6IP), syntenin (SDCBP) and syndecan-4 (SDC4) (Fig. 2a). Indeed, these three proteins were found among the most enriched in MBR+ compared to the other three fractions (Fig. 1e, highlighted in red) and are known to form a tripartite complex in interphase in the context of exosome formation[41]. Whether the same complex could be involved in cell division was unknown and could potentially reinforce the idea that exosome formation and cytokinetic abscission share common basic mechanisms. In addition, the implication of ALIX in abscission has been established (e.g., ref. [29]) but its exact role remains elusive.

**ALIX, syntenin, and syndecan-4 colocalize with ESCRT-III.** In fixed cells, endogenous ALIX and CHMP4B colocalized as two parallel stripes at the midbody (hereafter figured with white brackets) in bridges without observable abscission sites, as expected (Supplementary Fig. 4a). When bridges mature, the future abscission site also known as secondary ingression site (hereafter pointed with a white arrowhead) appears at the level of pinched and/or interrupted tubulin staining on one side of the midbody, typically 1–2 μm away, as previously characterized[12,39]. At this late stage, CHMP4B staining extends from the midbody to the abscission site, often appearing as a cone-shaped structure on one side of the midbody, as previously reported[12,13,20,21,25,30,31,39] (Fig. 2b). We found that endogenous ALIX colocalized with CHMP4B both at the midbody and at the abscission site (Fig. 2b). To our knowledge, ALIX had not been described at the abscission

site. Time-lapse spinning-disk confocal microscopy actually confirmed that abscission occurred at the tip of the ALIX cone (Supplementary Fig. 4b, arrow and Supplementary Movie 1). Similarly, we observed colocalization at the confocal microscopy level between ALIX/syntenin and syntenin/syndecan-4, both at the midbody and at the abscission site (Fig. 2b), using antibodies recognizing endogenous proteins (staining specificity confirmed in Supplementary Fig. 4c). Interestingly, super-resolution microscopy using structured illumination (SIM) showed that both syntenin and ALIX colocalized with CHMP4B at the outer rim of CHMP4B staining (Fig. 2c), a relative localization consistent with the molecular scheme presented in Fig. 2a and with the model presented in the final figure. Time-lapse spinning-disk confocal microscopy further revealed a striking time-dependent and concomitant enrichment of these proteins, first at the midbody, then at the abscission site before the cut (Fig. 2d; Supplementary Movies 2–4). Thus syndecan-4, syntenin, ALIX, and CHMP4B extensively and dynamically colocalized during the terminal steps of cytokinesis, notably at the abscission site.

**Mechanism of syndecan-4 and syntenin recruitment by ALIX.** We next investigated how syntenin and syndecan-4 are recruited to the intercellular bridge during cytokinesis. First, siRNA-mediated depletion of ALIX strongly reduced the proportion of bridges positive for endogenous syntenin (Fig. 3a, b). In contrast, syntenin depletion had no effect on ALIX recruitment, and syndecan-4 depletion had basically no impact on syntenin or ALIX recruitment (Fig. 3b; Supplementary Fig. 4d). Importantly, upon reintroduction in ALIX-depleted cells, the ALIX F676D mutant that cannot bind to syntenin[41,49] was unable to restore the recruitment of endogenous syntenin to the bridge, whereas wild-type ALIX could (Fig. 3c). Furthermore, a green fluorescent protein (GFP)-tagged syntenin ΔALIX (a triple mutant LYP-LAA that cannot interact with ALIX[41]) was no longer recruited to the bridge, whereas GFP-syntenin wild-type and GFP-syntenin ΔSDC (harboring point mutations in PDZ1 and PDZ2 that disrupt the interaction with syndecans[41,50]) were recruited (Fig. 3d). Thus, ALIX recruits syntenin to the intercellular bridge and this requires a direct interaction between ALIX and syntenin.

We next observed that syndecan-4 failed to be correctly recruited at bridges upon ALIX depletion (consistent with the results above) or syntenin depletion (Fig. 3e). Furthermore, GFP-syndecan-4 Δsynt (a C-terminal YA motif deletion mutant that is essential for syntenin binding[51]) was no longer recruited to the bridge (Fig. 3f). Of note, the GFP-syndecan-4 ΔECD (a mutant that lacks the entire extracellular domain but retains its

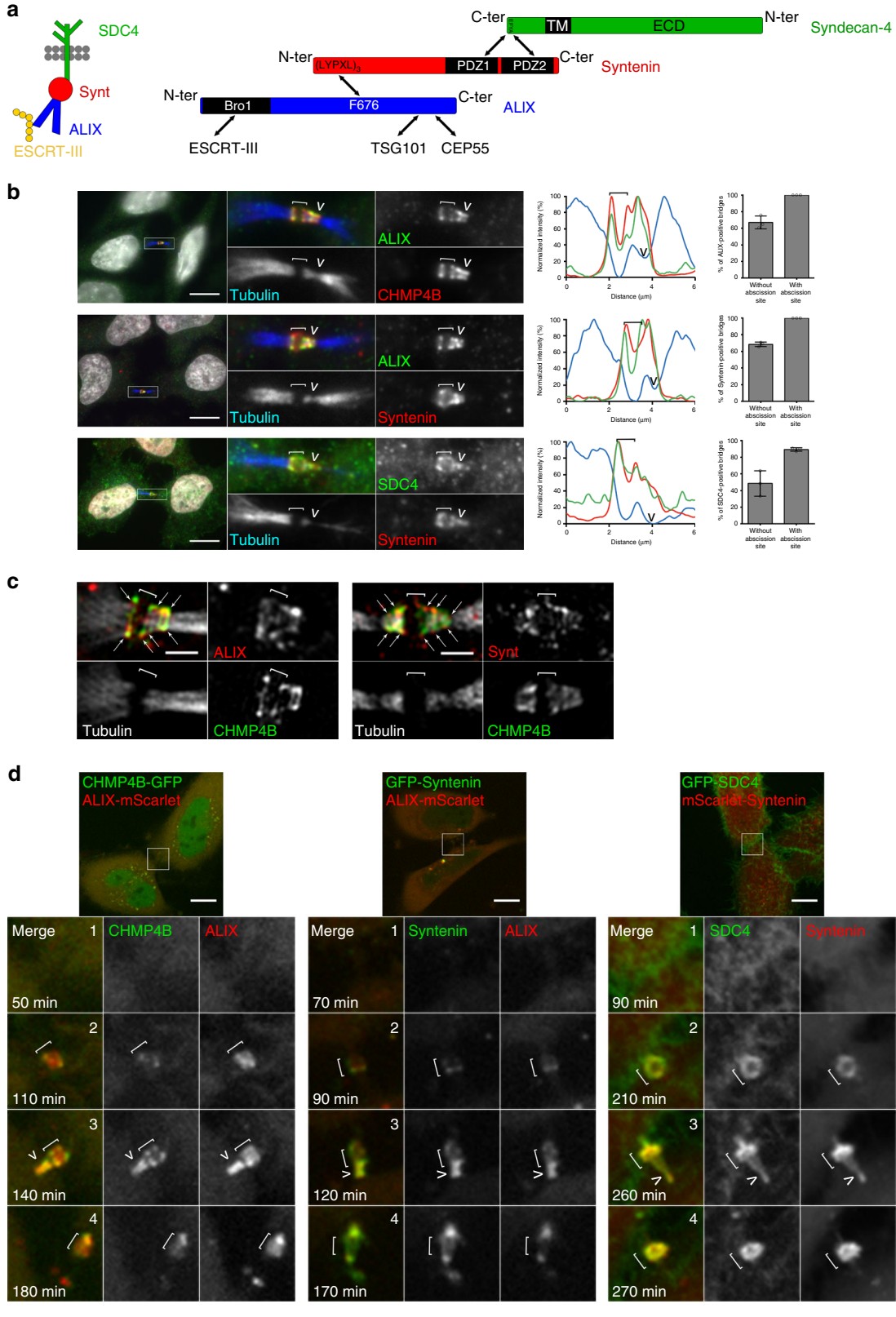

transmembrane and cytosolic domains) was properly recruited, indicating that the extracellular domain is dispensable for syndecan-4 localization at the midbody or at the abscission site (Fig. 3f). These results indicate that syntenin and its interaction with syndecan-4 are necessary for syndecan-4 localization at the bridge.

Altogether, we conclude that ALIX recruits syntenin that, in turn, recruits syndecan-4 at the intercellular bridge through direct interactions.

**Efficient abscission requires Syndecan-4, syntenin, and ALIX.** Functionally, depletion of either ALIX, syntenin or syndecan-4

**Fig. 2 Syndecan-4, syntenin, ALIX, and CHMP4B colocalize and are highly enriched first at the midbody then at the abscission site. a** The ESCRT-III-ALIX-syntenin-syndecan-4 complex. **b** Left panels: endogenous localization of ALIX, CHMP4B, syntenin, syndecan-4 (SDC4), and acetylated-tubulin in late bridges displaying abscission site in fixed HeLa cells. The SDC4 antibody recognizes the ectodomain. Middle panels: intensity profiles along the bridge of the corresponding images with matched colors from left panels. Right panels: percentage (mean ± SD) of bridges without and with abscission sites (displaying a pinched tubulin staining on the midbody side) positive for either ALIX, syntenin and syndecan-4. $n \geq 20$ cells, $N = 3$ independent experiments. One-sided Student's $t$ tests. **c** Structured illuminated microscopy images of endogenous ALIX/CHMP4B (left) and syntenin/CHMP4B (right), along with acetylated-tubulin staining in late bridges displaying abscission site. Arrows point to ALIX or syntenin localization at the outer rim of the CHMP4B staining. **d** Snapshots of time-lapse, spinning-disk confocal microscopy movies of cells co-expressing either CHMP4B-GFP/ALIX-mScarlet, GFP-Syntenin/ALIX-mScarlet, or GFP-SDC4/mScarlet-Syntenin. Selected time points show cells (1) before the recruitment of the fluorescently-labeled proteins, (2) after their enrichment at the midbody, (3) after their appearance at the abscission site, and (4) after abscission. Time 0 corresponds to the time frame preceding furrow ingression. See also corresponding Supplementary Movies 2–4. **b**, **d**: Scale bars: 10 μm. **c** Scale bar: 1 μm. Brackets and arrowheads mark the midbody and the abscission site, respectively.

led to a modest but reproducible increase of binucleated cells (Supplementary Fig. 5a). Nevertheless, time-lapse microscopy demonstrated that abscission was delayed in ALIX-depleted cells (Fig. 4a; Supplementary Fig. 5b), as previously reported, indicating that ALIX plays a specific role in abscission. Interestingly, abscission was also delayed after either syntenin or syndecan-4 depletions (Fig. 4b, c; Supplementary Fig. 5c, d). Abscission was fully restored by reintroducing either wild-type ALIX, syntenin or syndecan-4, respectively, ruling out off-target effects (Fig. 4a–c). Importantly, the ALIX F676D was unable to rescue the abscission defects due to ALIX depletion (Fig. 4a). Similarly, syntenin ΔALIX or syntenin ΔSDC failed to rescue the abscission defects observed after syntenin depletion (Fig. 4b). We conclude that ALIX, syntenin and syndecan-4 are important for normal abscission, and that the direct interactions between ALIX/syntenin on the one hand and syntenin/syndecan on the other hand are critical.

**Syndecan-4/syntenin and TSG101 cooperate for abscission.** TSG101 and ALIX were proposed to act in parallel for promoting abscission[29], and we confirmed that it is indeed the case, since co-depletion of TSG101 and ALIX led to a stronger delay in abscission (Fig. 4d, e, h). Interestingly, co-depleting either TSG101 and syntenin, or TSG101 and syndecan-4 also strongly increased the abscission delay, as compared to individual depletions (Fig. 4d, f–h). These results indicate that syndecan-4 and syntenin functionally cooperate with TSG101 for abscission. They also suggest that ALIX /syntenin/syndecan-4 and TSG101 act through distinct pathways to promote abscission.

**ESCRT-III at the abscission site rely on syndecan-4/syntenin.** We then investigated why abscission was delayed after ALIX, syntenin or syndecan-4 depletion. We first quantified the proportion of intercellular bridges with no ESCRT-III at all (early bridges), with ESCRT-III localized only at the midbody (bridges without secondary ingression) and with ESCRT-III both at the midbody and at the abscission site (bridges with constricted/interrupted tubulin staining, as in Fig. 2b). Depletion of either ALIX, syntenin or syndecan-4 considerably reduced the number of bridges with CHMP4B at the midbody + abscission site (Fig. 5a). This is consistent with the observed abscission delay in depleted cells (Fig. 4a–c). Importantly, neither ALIX, syntenin nor syndecan-4 were individually required for correct ESCRT-III recruitment at the midbody itself (the proportion of bridges stuck at this earlier stage was actually increased) (Fig. 5a). As expected, the localization of ESCRT-III at the abscission site was restored in ALIX-, syntenin-, or syndecan-4-depleted cells upon expression of siRNA-resistant versions of the corresponding wild-type proteins (Fig. 5b–d).

Importantly, the normal localization of ESCRT-III at the abscission site could not be restored by either ALIX F676D

(Fig. 5b), syntenin ΔALIX (Fig. 5c), syntenin ΔSDC (Fig. 5c), or syndecan-4 Δsynt (Fig. 5d). In contrast, syndecan-4 ΔECD localized at the bridge and behaved similarly to the wild type for ESCRT-III recruitment (Fig. 5d).

Confirming previous reports[29], we also observed that the presence of both TSG101 and ALIX is required for the recruitment of CHMP4B at the midbody itself (Fig. 5e). Strikingly, the depletion of both TSG101 and either syntenin or syndecan-4 essentially abolished the localization of CHMP4B at the abscission site (Fig. 5e).

Altogether, we conclude that direct interactions between ALIX and syntenin on the one hand, and syntenin and syndecan-4 on the other hand are critical for the recruitment of ESCRT-III specifically at the abscission site but not at the midbody itself. The defects of CHMP4B localization at the abscission site, which were aggravated when TSG101 was depleted, are consistent with the delay in abscission observed after ALIX, syntenin and syndecan-4 depletion (Fig. 4).

**Syndecan-4/syntenin/ALIX stabilize ESCRTs at the abscission site.** Finally, we investigated why ESCRT-III did not accumulate normally at the abscission site when either ALIX, syntenin or syndecan-4 were depleted, using fluorescent time-lapse microscopy in cells expressing a functional CHMP4B-GFP[31].

In control cells, CHMP4B-GFP accumulated first at the midbody then appeared on its side as a strong, large, cone-like signal pointing toward the abscission site (Fig. 5f; Supplementary Movie 5). Importantly, in ALIX-depleted cells, CHMP4B-GFP signal on the side of the midbody was less prominent and frequently disappeared as if it was unstable, something that we never observed in controls (Fig. 5g, yellow and cyan arrows highlight these transient pools of CHMP4B). As a consequence, the time between the first occurrence of CHMP4B on the side of the midbody and actual abscission was delayed (Supplementary Fig. 5e, up), and abscission eventually occurred without large, cone-shaped concentration of CHMP4B at the abscission site (Fig. 5g; Supplementary Movie 6). In syntenin- or syndecan-depleted cells, we again observed a transient and unstable localization of CHMP4B-GFP on the side of the midbody (Fig. 5h, i, yellow and cyan arrows; Supplementary Movies 7 and 8). Abnormal behavior of CHMP4B at the future abscission site was thus observed upon either ALIX, syntenin or syndecan-4 depletion (Fig. 5j). This was accompanied by a delay between the accumulation of CHMP4B at the midbody and abscission (Supplementary Fig. 5e, bottom), and CHMP4B-GFP signal on the side of the midbody took longer to appear (Supplementary Fig. 5e, middle). As a consequence, the time between the first occurrence of CHMP4B on the midbody side and actual abscission was delayed (Supplementary Fig. 5e, up). Although the phenotypes were similar, all the measured parameters were more impaired in ALIX-depleted cells as compared with

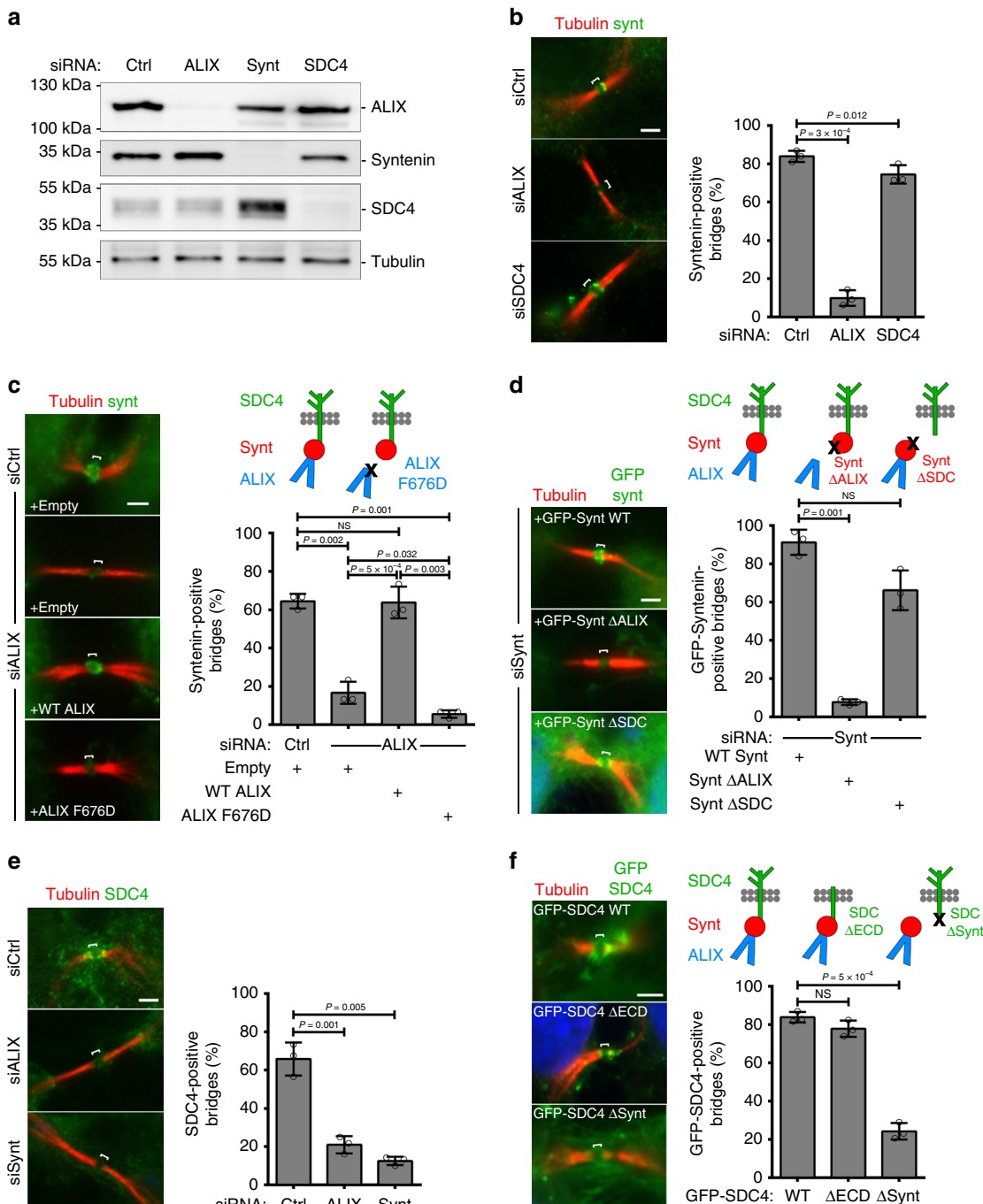

**Fig. 3 ALIX directly recruits syntenin and syntenin directly recruits syndecan-4 to the cytokinetic bridge. a** Western blots of protein extracts from cells treated with control, ALIX, syntenin (Synt), or syndecan-4 (SDC4) siRNAs revealed with the indicated antibodies. Loading control: β-tubulin. **b** Left panels: representative intercellular bridges from control, ALIX or syndecan-4 siRNA-treated cells stained for acetylated-tubulin and endogenous syntenin, as indicated. Right panel: percentage (mean ± SD) of bridges positive for syntenin after control, ALIX and syndecan-4 depletion. $n = 31–53$ cells, $N = 3$ independent experiments. (**c**) Left panels: representative intercellular bridges from control- or ALIX-depleted cells expressing either control (Empty), wild-type ALIX or ALIX F676D mutant (unable to interact with syntenin), and stained for acetylated-tubulin and endogenous syntenin, as indicated. Right panel: percentage (mean ± SD) of bridges with syntenin recruitment in the corresponding conditions. $n = 25–31$ cells, $N = 3$ independent experiments. **d** Left panels: representative intercellular bridges from syntenin-depleted cells expressing either GFP-syntenin wild-type, GFP-syntenin ΔALIX (unable to interact with ALIX) or GFP-syntenin ΔSDC (unable to interact with syndecan-4). Acetylated-tubulin and GFP signals are shown. Right panel: percentage (mean ± SD) of bridges with GFP-tagged syntenin recruitment in the corresponding conditions. $n = 14–33$ cells, $N = 4$ independent experiments. **e** Left panels: representative intercellular bridges from control, ALIX or syntenin siRNA-treated cells and stained for acetylated-tubulin and endogenous syndecan-4, as indicated. Right panel: percentage (mean ± SD) of bridges positive for syndecan-4 after control, ALIX and syntenin depletion. $n = 30–40$ cells, $N = 3$ independent experiments. **f** Left panels: representative intercellular bridges from cells expressing either GFP-syndecan-4 wild-type, GFP-syndecan-4 ΔECD (deleted from its entire extracellular domain) or GFP-syndecan-4 ΔSynt (unable to interact with syntenin). Acetylated-tubulin and GFP signals are shown. Right panel: percentage (mean ± SD) of bridges with GFP-tagged syndecan-4 recruitment in the corresponding conditions. $n = 23–41$ cells, $N = 3$ independent experiments. **b–f** Scale bars: 2 μm. Brackets mark the midbody. Panels **b–f**: one-sided Student's $t$ tests. NS non significant.

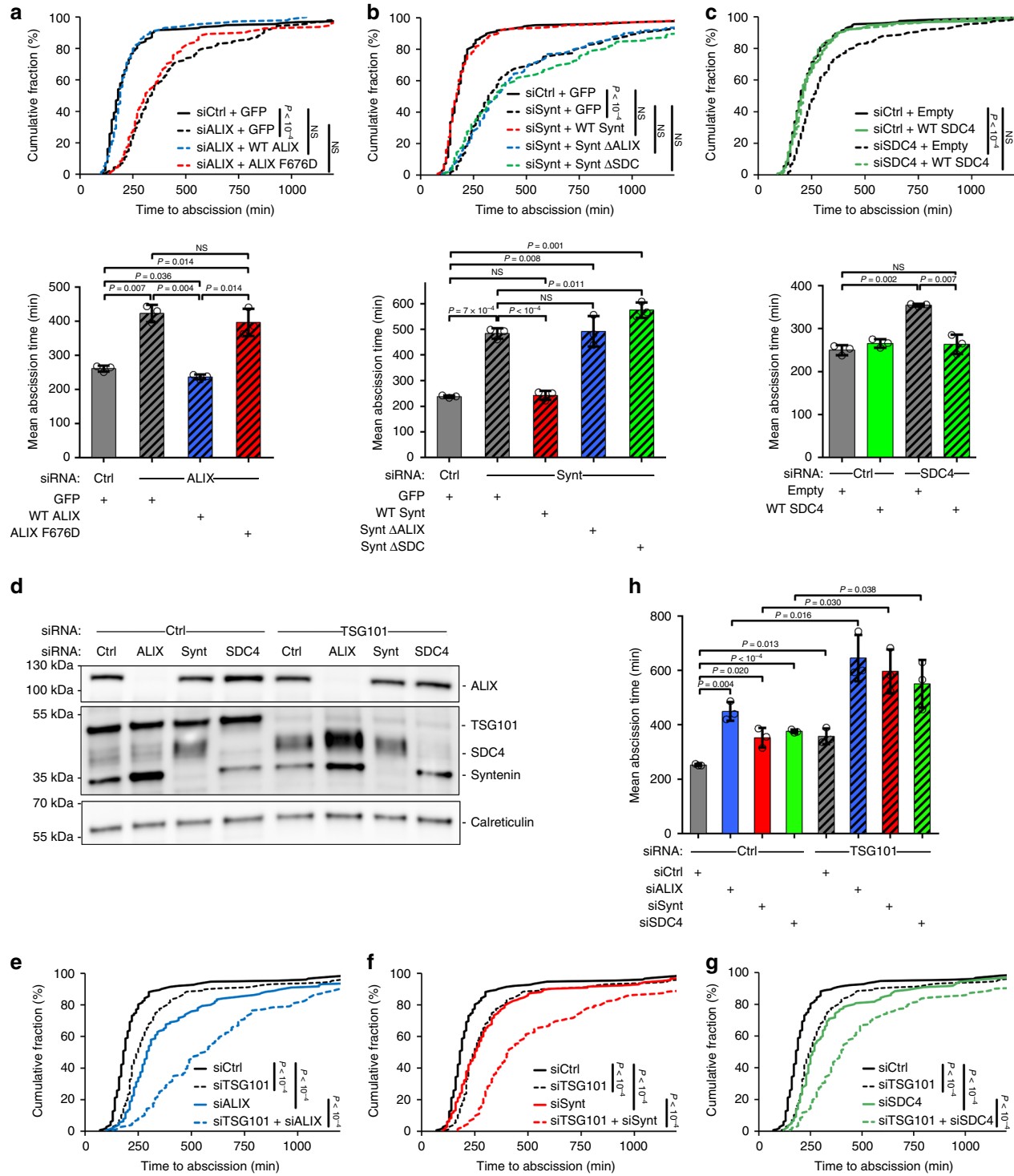

syntenin- or syndecan-4-depleted cells (Fig. 5j; Supplementary Fig. 5e), perhaps because ALIX directly recruits ESCRT-III and/or possible alternative pathways (such as TSG101) that can partially compensate when syntenin or syndecan-4 are depleted. Altogether, we conclude that syndecan-4-syntenin-ALIX promotes stable recruitment of ESCRT-III at the abscission site and is thus key for efficient cytokinetic abscission.

## Discussion
Most proteins involved in cytokinetic abscission strongly accumulate at the midbody[3,9]. Here, we identified 489 proteins enriched in purified MBR (*Enriched Flemmingsome*) (Fig. 1, Supplementary Data 1). To isolate MBRs, we developed an original flow cytometry-based protocol that yielded preparations that display three important features. First, the MBR+ fractions were highly pure. Second, the MBRs were obtained from unperturbed cells (no drugs for cell synchronization and no treatment for stabilizing actin or microtubules). Most importantly this purification did not involve any detergents. This allowed us to identify both transmembrane (29 proteins, Supplementary Data 1, TAB2, in blue) and membrane associated proteins in this organelle. This includes the 4 Rab proteins already involved in cytokinesis (Rab8, Rab11, Rab14, and Rab35)[3], as well as

**Fig. 4 ALIX, syntenin, and syndecan-4 are required for successful abscission. a** Transduced-HeLa cell lines expressing either GFP, wild-type ALIX-GFP or ALIX F676D-GFP were treated with either control or ALIX siRNAs. Abscission time (from furrow onset to abscission) was determined by phase-contrast time-lapse microscopy. Cumulative plot of the abscission times (upper panels) and mean abscission times (means ± SD) (lower panels) are shown. $n =$ 111–156 cells, $N = 3$ independent experiments. **b** Abscission time was determined as above in transduced cell lines expressing either GFP, wild-type mCherry-syntenin, mCherry-syntenin ΔALIX, or mCherry-syntenin ΔSDC and treated with either control or syntenin siRNAs, as indicated. $n = 105$–147 cells, $N = 3$ independent experiments. **c** Abscission time was determined as above in HeLa cells treated with either control or syndecan-4 siRNAs, and transfected with either empty plasmid or plasmid expressing siRNA resistant transcript encoding syndecan-4. $n = 130$–179 cells, $N = 3$ independent experiments. Time axis were stopped at 1200 min. **d** Western blots of protein extracts from cells treated with control, ALIX, syntenin (Synt), syndecan-4 (SDC4), TSG101 siRNAs, or a combination of these siRNAs and revealed with the indicated antibodies. Loading control: calreticulin. **e–g** Abscission time was determined in cells treated with either control, ALIX, syntenin, syndecan-4, TSG101 siRNAs, or a combination of these siRNAs. Mean abscission times (means ± SD) are depicted in (**h**). $n = 120$–152 cells, $N = 3$ independent experiments. All measurements were carried out in parallel but have been displayed in three graphs for clarity. Upper panels **a–c**, **e–g** (time distribution): nonparametric and distribution-free Kolmogorov–Smirnov KS tests. NS nonsignificant. Lower panels **a–c**, **h** (mean abscission times): one-sided Student's *t* tests.

---

transmembrane proteins such as chloride channels (CLIC1 and CLIC4, previously localized at the midbody[52]), adhesion/tethering/signaling molecules (ITGA3, PlexinB2, ICAM1, BST2, and CD44) and tetraspanins (CD9, CD9-P1, and TM4SF1) whose potential functions during cytokinesis will be studied in the future. These three points are key improvements, when comparing with previous proteomes of intercellular bridges, which already proved to be seminal in identifying essential proteins in cytokinesis[44,53]. Of note, 68 and 29% of the final list of Skop et al. (160 proteins from CHO cells) were respectively present in our *Total Flemmingsome* (1732 proteins) and *Enriched Flemmingsome* (489 proteins) (Supplementary Data 1, TAB6). The difference in extent of protein recovery between both studies could be explained by (1) differences in cell origins (hamster CHO vs. human HeLa), (2) differences in actual organelles (intercellular bridges before abscission vs. post-abscission, free midbodies as generated at the time of abscission), (3) the membrane integrity of the organelles (use of detergents vs. detergent-free, thus preserving cytosolic components), (4) reduced contaminations and finally the relative quantification of protein abundance (the *Enriched Flemmingsome* is based on significant enrichment compared to control fractions, including total cell lysates). While our paper was in revision, a noncomparative proteomic analysis of a MBR preparation, based on isolation from the culture media of HeLa cells by differential centrifugations associated with sucrose-gradient (no replica mentioned) has been published[54]. Interestingly, it showed 64% overlap with our *Total Flemmingsome* (558 out of 871 hits[54] are found in the *Total Flemmingsome*).

Our quantitative *Flemmingsome* was able to identify many *bona fide* cytosolic (ESCRT-related, actin-related, microtubule-related) and membrane/vesicle associated proteins (e.g., Rab11, Rab35, Rab8, and Rab14) involved in cytokinesis[3] but undetected in previous proteomic analysis (Fig. 1, Supplementary Data 1). Intriguingly, GO-term analysis indicated that ribosomal proteins were found enriched in the *Flemmingsome*, suggesting a functional interplay between cytokinesis and translation, as anticipated by studies in Drosophila[55,56]. In addition, proteasome inhibition by MG132 has been previously found to delay abscission[57] and here we revealed the presence of major proteasome components in the *Enriched Flemmingsome* (Fig. 1, GO-enrichment and Supplementary Data 1).

Importantly, 31% of the proteins present in the *Enriched Flemmingsome* (150 proteins) were already demonstrated to be localized to the furrow, bridge or midbody during cytokinesis and/or to be involved in cytokinesis (Supplementary Data 1, TAB2). This asserts the strength of our proteomic study, as well as its potential for identifying additional candidates (339) important for cytokinesis. To define the *Enriched Flemmingsome*, we decided to arbitrarily put the threshold of enrichment to 1.3-

fold and a 5% FDR. This allows to include proteins that are significantly enriched in MBRs and that are also present in the total cell lysate, as many proteins involved in cytokinesis (e.g. actin-related and traffic-related) have also functions in interphase. The *Flemmingsome* thus represents a useful resource for the cytokinesis community and we created a website to share this data (https://flemmingsome.pasteur.cloud/). Each hit in the *Enriched Flemmingsome*, in particular, was browsed in the literature for an eventual function/localization linked to cytokinesis and this reference database will be regularly updated.

Following up on the *Enriched Flemmingsome*, we found that ALIX, syntenin and syndecan-4 are indeed highly enriched at the midbody, then at the abscission site (Fig. 2). Functional analysis demonstrated that the three proteins are required for proper abscission (Fig. 4), and that they function together for ESCRT-III localization, specifically at the abscission site (Fig. 5). Of note, depletion of ALIX, syntenin or syndecan-4 revealed a specific role for these proteins in abscission rather than on intercellular bridge stability or integrity. Indeed, we observed only a modest, although reproducible, increase of binucleated cells (Supplementary Fig. 5a). For reasons that need further investigation but that we suspect to depend on differences in cell lines (e.g. HeLa ATCC vs. HeLa Kyoto), other studies reported a stronger increase in binucleated cells after ALIX depletion[21,22]. Nonetheless, our results indicate that ALIX, syntenin and syndecan-4 are important for promoting the coupling/interactions of the ESCRT-III machinery to the plasma membrane at the abscission site during the abscission step (see below) rather than for firmly attaching the plasma membrane to the midbody, the latter clearly depends on the physical interaction between MgcRacGAP and the membrane[58].

Interestingly, we previously reported that syntenin can directly bridge ALIX to syndecan-1/4 in vitro, and that ALIX-syntenin-syndecan are key for budding of intraluminal vesicles in MVBs and exosome production[41]. This likely depends on the ability of ALIX to recruit the ESCRT-III machinery at the neck of intraluminal vesicles in MVBs, but this could not be directly addressed given the small size of these necks. Here, we showed that the same module (ALIX-syntenin-syndecan-4) is used during cytokinesis at a much larger, micrometric scale, and found that it is actually critical for the stable association of ESCRT-III at the abscission site (Fig. 5). Of note, we did not detect any ALIX or syntenin at the plasma membrane in interphase cells, although syndecan-4 is clearly localized there (Supplementary Movie 4). This suggests that post-translational modifications of either ALIX, syntenin and/or syndecan-4, perhaps via phosphorylations[43,59] regulate the formation of the tripartite complex at the midbody and at the abscission site during cytokinesis.

Our results therefore reveal that ESCRT-III recruitment during cytokinesis relies on two successive, separable phases (Fig. 6).

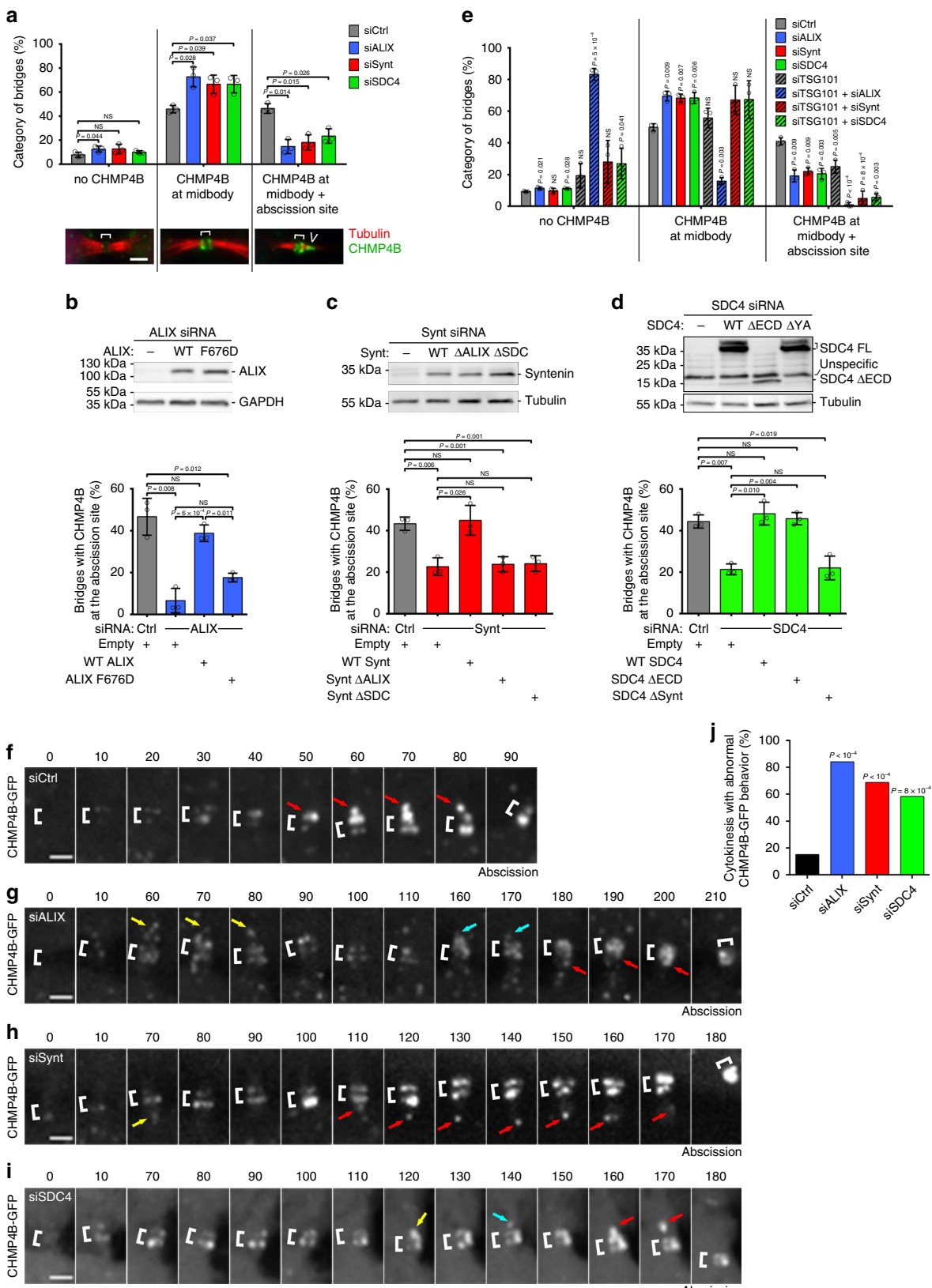

First, CEP55 directly interacts with and recruits both TSG101 and ALIX at the midbody[37] (Fig. 6a). At the midbody, ESCRT-III components are recruited in parallel, directly by ALIX and indirectly by ESCRT-I/II[29]. Accordingly, both TSG101 and ALIX must be simultaneously depleted to prevent ESCRT-III recruitment at the intercellular bridge[29]. However, CEP55 cannot account for ESCRT-III localization at the abscission site since it is absent from this location (Supplementary Fig. 5f). We now found that ALIX plays a key additional role in cytokinesis: it recruits syntenin, which in turn interacts with the transmembrane protein

**Fig. 5 Persistent recruitment of ESCRT-III to the abscission site depends on the syndecan4-syntenin-ALIX module. a** Cells treated with the indicated siRNAs were stained for endogenous CHMP4B and acetylated-tubulin. CHMP4B localization in late cytokinetic bridges was classified into three categories: (1) no staining, (2) CHMP4B localized only at the midbody, or (3) CHMP4B localized both at the midbody and at the abscission site (see representative images). The proportion of each category was quantified in control and depleted cells. $n = 49–83$ cells, $N = 3$ independent experiments. **b–d** Cells were depleted for either ALIX (**b**), syntenin (**c**), or syndecan-4 (**d**) and transfected with control plasmid (−) or with plasmids encoding either wild type or mutant versions of ALIX, syntenin, or syndecan-4. Upper panels: western blots were revealed with the indicated antibodies. Loading controls: GAPDH or β-tubulin. Lower panels: percentage (mean ± SD) of bridges with CHMP4B at the abscission site in each condition. **b** $n = 28–31$ cells; **c** $n = 32–103$ cells, **d** $n = 33–56$ cells. $N = 3$ independent experiments. **e** Cells were treated with control, ALIX, syntenin, syndecan-4, TSG101 siRNAs, alone or in combination. CHMP4B localization was quantified as in (**a**). $n = 26–66$ cells, $N = 3$ independent experiments. **f–i** HeLa cells stably expressing CHMP4B–GFP were treated with either control (**f**), ALIX (**g**), syntenin (**h**), or syndecan-4 (**i**) siRNAs and recorded by spinning-disk confocal time-lapse microscopy every 10 min. Zooms of the intercellular bridges are displayed. Time 0 corresponds to the frame preceding the arrival of CHMP4B at the midbody. Brackets mark the midbody. Arrows point toward pools of CHMP4B on the midbody side. Red arrows correspond to the CHMP4B leading to abscission (last time frame). Yellow and cyan arrows point to transient and unstable CHMP4B pools observed in depleted cells. See corresponding Supplementary Movies 5–8. **j** Quantification of cytokinesis with abnormal CHMP4B-GFP behavior (disappearance of signal on the side of the midbody and fragmented cones) for each condition reported in (**f–i**). $n = 24–38$ cells from 5 independent experiments. One-sided Fisher's exact tests. **a**, **f–i** Scale bars: 2 µm. Brackets mark midbodies in (**a**, **f–j**). Panels **a–e**: means ± SD and one-sided Student's $t$ tests. NS nonsignificant.

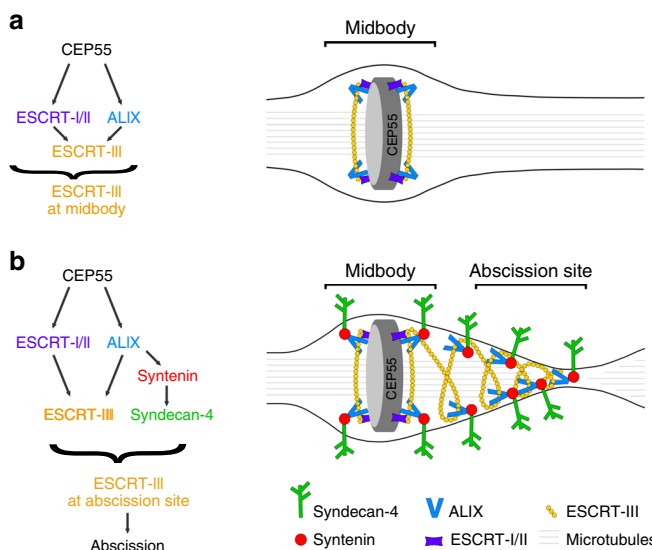

**Fig. 6 Working model: ALIX-syntenin-syndecan-4 couples the ESCRT-III machinery to the plasma membrane at the abscission site for efficient scission. a** ESCRT-III localization at the midbody depends on its recruitment by ALIX (blue) and ESCRT-I/-II (violet), which are targeted to the midbody by MKLP1-associated CEP55 (gray). This first step does not require syntenin or syndecan-4. **b** ESCRT-III localization at the abscission site, located on one side of the midbody, depends on the tripartite ALIX-syntenin-syndecan-4 module. ALIX-syntenin (red), by directly coupling ESCRT-III (yellow) on the one hand and the transmembrane protein syndecan-4 (green) on the other hand is proposed to help maintain ESCRT-III polymers at the abscission site until the final cut.

syndecan-4 (Fig. 6b). Importantly, the ALIX-syntenin-syndecan-4 module is required to stably maintain ESCRT-III components at the abscission site. In the absence of either ALIX, syntenin or syndecan-4, ESCRT-III components can polymerize and extend from the midbody toward the abscission site. However, ESCRT-III recruitment at the abscission site takes longer and is unstable, resulting in delayed abscission (Fig. 5). We thus propose that ALIX-syntenin, by physically coupling ESCRT-III on the one hand and the transmembrane protein syndecan-4 on the other hand, help to maintain ESCRT-III polymers at the abscission site until the final cut (Fig. 6b). Consistently, CHMP4B and syndecan-4 displayed correlated patterns at the abscission site in cells that were fixed shortly before the actual cut (Supplementary Fig. 5g). Furthermore, in syndecan-4-depleted cells, the

expression of syndecan-4 cytoplasmic tail alone was unable to rescue ESCRT-III localization at the abscission site (Supplementary Fig. 5h), indicating that both the transmembrane and the cytoplasmic tail of syndecan-4 are critical. In summary, the chain of interactions between ALIX/syntenin/syndecan-4 appears critical for proper ESCRT-III localization at this site and thus abscission (Figs. 4 and 5). Furthermore, this model explains previous observations showing that depleting ALIX alone, which has little effect on ESCRT-III recruitment at the midbody itself, indeed has an impact on abscission[29]. Interestingly, co-depletion of TSG101 and either syntenin or syndecan-4 did not abolish ESCRT-III recruitment at the midbody (ALIX is still present), but strongly impacted on ESCRT-III localization at the abscission site (Fig. 5e) and consequently on abscission (Fig. 4d–h). This suggests that ALIX and TSG101 actually cooperate for localizing ESCRT-III at the abscission site (where they both localize), and not only for recruiting ESCRT-III at the midbody. Future studies will be required to understand how ALIX (via syntenin/syndecan-4) and TSG101 (via an unknown mechanism) promote abscission through distinct pathways.

In other related ESCRT-dependent membrane scission events, such as exosome formation in MVBs and HIV budding, ALIX also plays a key role[4,7]. As mentioned above, we previously found that the ALIX-syntenin-syndecan module is essential for proper exosome scission[41]. TSG101 and ALIX cooperate and are both important in HIV budding, but TSG101 plays a more prominent role, in contrast to abscission (refs. [29] and Fig. 4). Remarkably, HIV appears to have hijacked and simplified this ALIX-syntenin-syndecan module, since the GAG protein (which is tightly associated to the plasma membrane through myristoylation) contains the ALIX-interacting LYPxL motif (that is found three times in syntenin), bypassing the need for syndecan-syntenin in the scission step[49,60–62]. Thus, our study suggests that the coupling of the ESCRT-III machinery to a membrane protein via ALIX-syntenin or equivalent modules represents a critical requirement for efficient scission during cytokinesis, exosome formation, and retroviral budding.

## Methods
**Cell cultures.** HeLa cells CCL-2 (ATCC) and HeLa GFP-MKLP2[45] were grown in Dulbecco's Modified Eagle Medium (DMEM) GlutaMax (31966; Gibco, Invitrogen Life Technologies) supplemented with 10% fetal bovine serum and 1× penicillin–streptomycin (Gibco) in 5% $CO_2$ at 37 °C. The HeLa GFP-MKLP2 cells were cultured in G418 (Gibco) after FACS selection. CHMP4B-GFP and GFP-syndecan-4 stable cell lines were generated by electroporating HeLa ATCC cells with the corresponding plasmids, followed by G418 selection (Gibco) and selected by FACS sorting. GFP, ALIX-LAP-GFP WT, ALIX-LAP-GFP F676D, mCherry-syntenin WT, mCherry-syntenin ΔALIX and mCherry-syntenin ΔSDC stable cell

lines were generated by lentiviral transduction of HeLa ATCC cl-2 cells and selected by FACS sorting.

**Transfections and siRNAs**. Plasmids were transfected in HeLa cells for 24 or 48 h using X-tremeGENE 9 DNA reagent (Roche). For silencing experiments, HeLa cells were transfected with 25 nM siRNAs for 48 h using HiPerFect (Qiagen) or Lipofectamine RNAiMAX (Invitrogen), following the manufacturer's instructions. siRNAs against Luciferase (used as control, 5′CGUACGCGGAAUACUUCGA3′), ALIX (5′CCUGGAUAAUGAUGAAGGA3′), Syntenin (5′GAAGGACUCUCAAA UUGCA3′), Syndecan-4 (5′GUGAGGAUGUGUCCAACAA3′) and TSG101 (5′CC UCCAGUCUUCUCUCGUC3′) have been synthetized by Sigma. In rescue experiments, cells were first transfected for 72 h with siRNAs using HiPerFect, then cotransfected by plasmids encoding untagged proteins and either GFP or H2B-GFP to detect transfected cells, using X-tremeGENE 9 DNA reagent for an additional 24 h. SiRNA-resistant versions of ALIX, syntenin and syndecan-4 have been obtained by mutating 6 bp of the siRNA-targeting sequence using NEBaseChanger (NEB).

**Plasmid constructs**. Human ALIX, syntenin, and syndecan-4 cDNAs were subcloned into Gateway pENTR plasmids and eGFP, mScarlet or untagged transient expression vectors were generated by LR recombination (Thermo Fisher). All point mutations have been generated using NEBaseChanger (NEB), including ALIX F676D, syntenin ΔALIX (Y4A, P5A, Y46A, P47A, Y50A, and P51A)[41], syntenin ΔSDC (K119A, S171H, D172E, K173Q, K203A, K250S, D251H, and S252E)[50] syndecan-4 ΔECD (deleted from E19 to E145 included), syndecan-4 Δsynt (deleted for the last two amino acids, Y197 and A198)[51]. GFP-syndecan-4 was constructed by fusing the eGFP sequence between the syndecan-4 M1–E145 and E142–A198 sequences.

**Western blots**. Western blot experiments comparing samples used in proteomic studies: Protein extracts from Total, MBRE, MBR+ and MBR− fractions were obtained directly after addition of 2% SDS to the samples. Proteins from approximately $2.5 \times 10^6$ MBR+ particles from flow cytometry were loaded after lyophilization and resuspended in Laemmli 1× loading buffer on a 4–12% gradient SDS-PAGE gel (Bio-Rad). Serial dilutions (2-fold) were carried on from 5 to 10 µg extracts from Tot and MBRE preparations for comparisons between the samples and SyproRuby protein blot stain (Bio-Rad) used to determine the protein concentrations in the different samples. Lanes with same levels of protein are shown in Fig. 1c and indicated as lane 1 and serial dilutions mentioned from this lane in Supplementary Fig. 1b. Uncropped blots are displayed in Supplementary Fig. 6.

Western blot experiments after siRNA treatment *were carried out as follows*: cells treated with siRNAs were lysed in NP-40 extract buffer (50 mM Tris, pH 8, 150 mM NaCl, 1% NP-40) containing protease inhibitors. Totally, 20 µg of lysate were migrated in 10% or 4–15% gradient SDS-PAGE gels (Bio-Rad Laboratories), transferred onto polyvinylidene fluoride membranes (Millipore) and incubated with corresponding antibodies in 5% milk in 50 mM Tris-HCl pH 8.0, 150 mM NaCl, 0.1% Tween20, followed by horseradish peroxidase-coupled secondary antibodies (1:20,000, Jackson ImmunoResearch) and revealed by chemiluminescence (GE Healthcare). For western blots against syndecan-4, cell extracts were treated with heparinase (AMS.HEP-ENZ III) and chondroitinase (AMS.E1028-02) for 3 h at 37 °C before migration. Uncropped western blots are displayed in Supplementary Fig. 7.

**Immunofluorescence and image acquisition**. HeLa cells were grown on coverslips and then fixed either with paraformaldehyde (PFA) 4% for 20 min at room temperature, with methanol for 3 min at −20 °C, or with trichloroacetic acid 10% for 20 min at room temperature. Cells were then permeabilized with 0.1% Triton-X100, blocked with PBS containing 0.2% bovine serum albumin (BSA) and successively incubated for 1 h at room temperature with primary (Supplementary Table 1) and secondary antibodies diluted in PBS containing 0.2% BSA with DAPI staining (0.5 mg/ml, Serva). Cells were mounted in Mowiol (Calbiochem). Images were acquired with an inverted TiE Nikon microscope, using a ×100 1.4 NA PL-APO objective lens or a ×60 1.4 NA PL-APO VC objective lens and MetaMorph software (MDS) 7.8.0.0 driving a CCD camera (Photometrics Coolsnap HQ). Images were then converted into 8-bit images using ImageJ software (NIH). Purified MBR+ from flow cytometry were concentrated at 1200*g* and a 5 µl-drop was incubated overnight on a glass coverslip. The MBR+ particles were processed for immunofluorescence as described above for cells. Cell Mask (C10045, Thermo-Fisher) staining was performed on the GFP-MKLP2 adherent cells as indicated by the manufacturer, and then the MBR+ particles purified by flow cytometry as described above.

**Structured illumination microscopy (SIM)**. SIM was performed on a Zeiss LSM 780 Elyra PS1 microscope (Carl Zeiss, Germany) using C Plan-Apochromat 63×/1.4 oil objective with a 1.518 refractive index oil (Carl Zeiss). The fluorescence signal is detected on an EMCCD Andor Ixon 887 1 K. Raw images are composed of fifteen images per plane per channel (five phases, three angles), and acquired with a Z-distance of 0.11 µm. Acquisition parameters were adapted from one image to one other to optimize the signal to noise ratio. SIM images were processed separately

for each channel with ZEN software and then corrected for chromatic aberration using 100-nm TetraSpeck microspheres (ThermoFisher Scientific) embedded in the same mounting media as the sample. The SIMcheck plugin in imageJ was used to analyze the quality of the acquisition and the processing in order to optimize parameters for resolution, signal-to-noise ratio, and reconstruction pattern[63].

**Time-lapse microscopy**. For time-lapse phase-contrast imaging, HeLa cells were plated on glass bottom 12-well plates (MatTek) and put in an open chamber (Life Imaging) equilibrated in 5% $CO_2$ and maintained at 37 °C. Time-lapse sequences were recorded every 10 min for 48 h using an inverted NikonEclipse TiE microscope with a ×20 0.45 NA Plan Fluor ELWD controlled by Metamorph software (Universal Imaging). For time-lapse fluorescent microscopy, images were acquired using an inverted Eclipse TiE Nikon microscope equipped with a CSU-X1 spinning disk confocal scanning unit (Yokogawa) and with a EMCCD Camera (Evolve 512 Delta, Photometrics). Images were acquired with a ×60 1.4 NA PL-APO VC and MetaMorph software (MDS).

**Statistics and reproducibility data**. All values are displayed as mean ± SD (standard deviation) for at least three independent experiments (as indicated in the figure legends). Significance was calculated using unpaired, one-sided *t* tests or one-sided exact Fisher's tests, as indicated. For comparing distribution of abscission times, a nonparametric Kolmogorov–Smirnov test was used. In all statistical tests $p > 0.05$ was considered as non significant. *p* values are indicated in the Figures.

Provided representative images have been observed reproducibly as indicated below: Fig. 1a ($n > 20$ cells), Fig. 1b ($n > 30$ FACS sortings), Fig. 1c ($n = 2$ blots), Fig. 1d ($n > 20$ midbodies), Fig. 2b ($n > 30$ ICBs), Fig. 2c ($n > 3$ ICBs), Fig. 2d ($n > 10$ movies), Fig. 3a ($n = 2$ blots), Fig. 4d ($n = 2$ blots), Supplementary Fig. 4a ($n > 30$ ICBs), Supplementary Fig. 4b ($n = 3$ movies), Supplementary Fig. 5b ($n > 400$ movies), Supplementary Fig. 5f ($n > 20$ cells), and Supplementary Fig. 5g ($n > 20$ cells).

**Sample preparation for mass spectrometry**. HeLa GFP-MKLP2[45] were detached from flasks with 0.05% trypsin diluted in 0.02% EDTA (25300; Gibco, Invitrogen Life Technologies) and plated at $8 \times 10^5$ cells/well on 10-cm dishes for 3 days. Cells were rinsed 3-times with HBSS and then incubated in 2 mM EDTA/HBSS for 10 min at 37 °C to detach MBRs from the cell surface. The "Total" fraction represented the whole fraction of detached cells including the EDTA-supernatant (SN). The cells were pelleted by centrifugation (5 min at 70*g*). The "MBRE" was adapted from ref. [14]: the SN from the first centrifugation was centrifuged again (5 min at 70*g*) and SN from this step was aliquoted to 300 µl for another centrifugation (10 min at 70*g*). The MBRs were concentrated from the last SN by 60 min centrifugation at 1200 *g*.

**Flow cytometry sorting**. MBRs were detached from HeLa GFP-MKLP2[45] cells with EDTA-treatment as described above. The SN from the first 70 *g* centrifugation was collected. Sorting of MBR+ and MBR− particles was performed on a BD Biosciences FACS ARIA III. Neutral Density filter 1.0 has been used to detect small particles. Totally, 65,000 particles were gated on a pseudo-color plot looking at GFP vs. SSC-A parameters, both in log scales, as indicated. Cells have been excluded from the sorting gates after analysis of an unstained cell suspension as control (Fig. 1b; Supplementary Fig. 1b). The MBR+ (GFP-positive population) and MBR− (GFP-negative counterpart) populations sorted by flow cytometry were concentrated by 60 min centrifugation at 1200*g* at 4 °C. The proteins from all the samples were solubilized in 2% SDS and further prepared for in-gel or in-solution digest.

**Preparation of samples for mass spectrometry**. In gel digestion: In-gel digestion was performed by standard procedures[64]. Proteins (10 µg) were loaded on a SDS-PAGE gel (4–20% gradient, Expedeon). The electrophoretic migration of the gel was stopped after the stacking and the gel was stained with Coomassie Blue (InstantBlue™, Expedeon) and each lane was cut into three gel bands. Gel slices were washed several times in 50 mM ammonium bicarbonate, acetonitrile (1:1) for 15 min at 37 °C. Disulfide bonds were reduced with 10 mM DTT and cysteine alkylated with 55 mM IAA. Trypsin (Promega) digestion was performed overnight at 37 °C in 50 mM ammonium bicarbonate. Peptides were extracted from the gel by two incubations in 10% formic acid, acetonitrile (1:1) for 15 min at 37 °C. Extracts were dried in a Speed-Vac, and resuspended in 2% acetonitrile, 0.1% formic acid prior to LC–mass spectrometry (MS)MS analysis. For each sample (Total, MBRE, MBR+, and MBR−) five independent preparations were run on SDS-PAGE; an experimental replicate was made as an internal control for the MBR+/MBR− samples (numbered 3 and 4, Supplementary Data 1, TAB3).

In-solution digestion *(eFASP)*: Protein samples extracted in SDS were digested using eFASP protocol[48]. Filter units and collection tubes were incubated overnight in passivation solution: 5% (v/v) TWEEN®-20. All buffer exchanges were carried out by centrifugation at 14,000 *g* for 10 min. Briefly, 10 µg of proteins from each sample were transferred into 30,000 MWCO centrifugal unit (Microcon® Centrifugal Filters, Merck) completed to 200 µL with exchange buffer (8 M urea, 0.2% DCA, 100 mM ammonium bicarbonate pH 8). Disulfide bonds were reduced with 5 mM TCEP (Sigma) for 1 h. Proteins were buffer-exchanged with three

rounds of 200 μL of exchange buffer. Buffer was then exchanged for an alkylation buffer (50 mM iodoacetamide, Urea 8 M, 100 mM ammonium bicarbonate pH 8) in the dark for 1 h. One wash with 200 μL of exchange buffer was done to remove the alkylating agent, followed by three buffer exchanges with 200 μL of digestion buffer (0.2% DCA/50 mM ammonium bicarbonate buffer pH 8). Totally, 100 μL of digestion buffer containing 1:50 ratio of sequencing-grade modified trypsin (Promega) per amount of protein was added to the retentate. Proteolysis was carried out at 37 °C overnight. Three rounds of 50 μL of recovery buffer (50 mM ammonium bicarbonate pH 8) were used to elute the peptide-rich solution. Then peptides were processed as described in ref. [48] and resuspended in 2% acetonitrile, 0.1% formic acid prior to LC–MS/MS analysis. For eFASP, three independent replicates were made for Tot and MBRE and two independent replicates for MBR+ and matched MBR− flow cytometry-sorted samples (Supplementary Data 1, TAB3).

**MS analysis.** Tryptic peptides from in-gel digestion were analyzed on a Q Exactive HF instrument (Thermo Fisher Scientific, Bremen) coupled with an EASY nLC 1000 chromatography system (Thermo Fisher Scientific). Sample was loaded on an in-house packed 50 cm nano-HPLC column (75 μm inner diameter) with C18 resin (1.9 μm particles, 100 Å pore size, Reprosil-Pur Basic C18-HD resin, Dr. Maisch GmbH, Ammerbuch-Entringen, Germany) after an equilibration step in 100% solvent A ($H_2O$, 0.1% FA). Peptides were first eluted using a 2–7% gradient of solvent B (ACN, 0.1% FA) during 5 min, then a 7–23% gradient of solvent B during 80 min, a 23–45% gradient of solvent B during 40 min and finally a 45–80% gradient of solvent B during 5 min all at 250 nL min$^{-1}$ flow rate. The instrument method for the Q Exactive HF was set up in the data dependent acquisition mode. After a survey scan in the Orbitrap (resolution 60,000), the 10 most intense precursor ions were selected for HCD fragmentation with a normalized collision energy set up to 28. Charge state screening was enabled, and precursors with unknown charge state or a charge state of 1, 7, 8, and >8 were excluded. Dynamic exclusion was enabled for 45 s.

Tryptic peptides from eFASP digestion were analyzed on a Q Exactive plus instrument (Thermo Fisher Scientific, Bremen) coupled with an EASY nLC 1000 chromatography system (Thermo Fisher Scientific) and processed as described above. Peptides were first eluted using a 2–5 % gradient of solvent B (ACN, 0.1% FA) during 5 min, then a 5–22% gradient of solvent B during 150 min, a 22–45% gradient of solvent B during 60 min and finally a 45–80% gradient of solvent B during 10 min all at 250 nL min$^{-1}$ flow rate. The instrument method for the Q Exactive Plus was set up in the data dependent acquisition mode. After a survey scan in the Orbitrap (resolution 70,000), the 10 most intense precursor ions were selected for HCD fragmentation with a normalized collision energy set up to 28. Charge state screening was enabled, and precursors with unknown charge state or a charge state of 1, 7, 8, and >8 were excluded. Dynamic exclusion was enabled for 45 s.

**Protein identification and quantification.** All data were searched using Andromeda[65] against a Human Uniprot database (downloaded in 2015 08 18, 20,204 entries), usual known MS contaminants and reversed sequences of all entries. Andromeda searches were performed choosing trypsin as specific enzyme with a maximum number of two missed cleavages. Possible modifications included carbamidomethylation (Cys, fixed), oxidation (Met, variable), and Nter acetylation (variable). The mass tolerance in MS was set to 20 ppm for the first search then 4.5 ppm for the main search and 20 ppm for the MS/MS. Maximum peptide charge was set to seven and five amino acids were required as minimum peptide length. The "match between runs" feature was applied for samples having the same experimental condition with a maximal retention time window of 1 min. One unique peptide to the protein group was required for the protein identification. A FDR cutoff of 1% was applied at the peptide and protein levels. Quantification was performed using the XIC-based LFQ algorithm with the Fast LFQ mode as described in ref. [66]. Unique and razor peptides, included modified peptides, with at least two ratio count were used for quantification.

**Statistical analysis for proteomic studies.** For the differential analyses, proteins identified in the reverse and contaminant databases and proteins "only identified by site" were first discarded from the list of identified proteins. Then, proteins exhibiting fewer than two quantified values in at least one condition were discarded from the list. After log2 transformation of the leftover proteins, LFQ values were normalized by median centering within conditions (*normalized* function of the R package *DAPAR*[67]). Remaining proteins without any LFQ value in one of both conditions have been considered as proteins quantitatively present in a condition and absent in another (Supplementary Data 1, TAB4). They have therefore been set aside and considered as differentially abundant proteins. Next, missing values were imputed using the imp.norm function of the R package norm[68]. Proteins with a fold-change under 1.3 have been considered not significantly differentially abundant. Statistical testing of the remaining proteins (having a fold-change over 1.3) was conducted using a *limma t* test[69] thanks to the R package *limma*[70]. An adaptive Benjamini–Hochberg procedure was applied on the resulting *p* values thanks to the function *adjust.p* of R package *cp4p*[71] using the robust method described in ref. [72] to estimate the proportion of true null hypotheses among the set of statistical tests.

The proteins associated to an adjusted *p* value inferior to a FDR of 5% have been considered as significantly differentially abundant proteins. Finally, the proteins of interest are therefore those which emerge from this statistical analysis supplemented by those which are considered to be absent from one condition and present in another. Results of these differential analyses are summarized in Supplementary Fig. 2 and Supplementary Data 1. The merged volcano plot (Fig. 1e, upper panel) is a summary of the six comparisons MBR+ vs. control (MBR−, MBRE, or Total; using either eFASP or Gel). The *x*-axis represents the maximum log2 fold-change between MBR+ and the different controls. The "*merged p value*" (*y*-axis) has been obtained using the Fisher's method from the different *p* values that have been measured in the comparisons. Note that these two quantities have been computed only when data were available.

**UpsetR graph and STRING functional association network.** The Upset graph (Supplementary Fig. 1d) represents the distribution of the significant proteins coming from the different statistical analyzes. The Venn diagram represents the numbers of differentially abundant proteins in function of the kind of protein sample preparation (In-Gel or eFASP). Functional association network was determined by STRING[73] and displayed using Cytoscape[74].

**GO cluster diagrams.** The over-representation analyses of GO-term clusters in Fig. 1g and Supplementary Fig. 1e have been performed using hypergeometric tests. GO-terms have been grouped following 11 different functional categories (details in Supplementary Table 1, TAB7). Hypergeometric tests were performed to test the overrepresentation hypothesis for each cluster using R software. All the proteins identified by MS have been used as background for the hypergeometric tests. A significantly low *p* value means the proportion of proteins related to a GO-term cluster is significantly superior in the considered list [Total Flemmingsome (Fig. 1g) or proteins more abundant in MBR+ than MBRE (Supplementary Fig. 1e)] than in this background.

**Reporting summary.** Further information on research design is available in the Nature Research Reporting Summary linked to this article.

## Data availability

The mass spectrometry proteomics data have been deposited to the ProteomeXchange Consortium via the PRIDE[75] partner repository with the dataset identifier PXD013219. The Flemmingsome Website: https://flemmingsome.pasteur.cloud/
All material described in this paper will be made available to readers and be sent upon request.

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

## Acknowledgements

We thank R. Basto, G. Hickson, J. Mathieu, J.-R. Huyhn, R. Shaughnessy, M. Serres, and T. Wai for critical reading of the paper; the Echard Lab members for helpful discussions; the Recombinant antibodies platform (TAb-IP, Institut Curie, Paris) and the DSHB (University of Iowa) for antibodies. GFP-MKLP2 cells were from the Hyman Lab MPI-MCBG Dresden[45]. UTechS PBI is part of the France–BioImaging infrastructure network (FBI) supported by the French National Research Agency (ANR-10-INSB-04; Investments for the Future), and acknowledges support from ANR/FBI and the Région Ile-de-France (program "Domaine d'Intérêt Majeur-Malinf") for the use of the Zeiss LSM 780 Elyra PS1 microscope. We thank P.H. Commere from the Utechs CB, Institut Pasteur for FACS sorting. This work has been supported by Institut Pasteur, CNRS, and ANR (AbCyStem, Cytosign) to A.E. C.A. received a fellowship from the Doctoral School Complexité du Vivant ED515, contrat n° 2412/2016 and AMX. A.P. received a fellowship from the Doctoral School Complexité du Vivant ED515, contrat n°2611 bis/2016 and Fondation ARC pour la recherche sur le cancer (DOC20190508876).

## Author contributions

C.A. carried out and analyzed the experiments presented in Figs. 2–5 and Supplementary Fig. 4 and 5; N.G.R. and A.P. in Fig. 1a–d and Supplementary Fig. 1b, c; S.F. in Fig. 1d and Supplementary Fig. 1a; F.M. in Fig. 3c; N.GR. and S.S. setup the flow cytometry-purification protocol for Fig. 1b, Supplementary Fig. 1b; N.G.R. and A.P. setup MBRE protocols. F.C. and M.R. assisted with technical help. T.D, M.D., J.C.R and M.M. designed and carried out the mass spectrometry studies; Q.G.G. did statistical analyses and Fig. 1e–g, Supplementary Fig. 1d, 1e, 2, and 3; N.G.R., Q.G.G., M.M., M.D., J.C.R. and T.D. contributed to the Supplementary Data 1; H.M. created the website. We acknowledge the help of Thomas Menard from the IT Department at the Institut Pasteur for this work. A.S. carried out and analyzed the experiments presented in Fig. 2c. P.Z. provided reagents and helpful discussions. A.E. conceived the project and secured funding. N.G.R. and A.E. supervised the proteomic data; A.E. supervised the other data. A.E. wrote the paper with the help of C.A., N.G.R., M.D., T.D., Q.G.G., J.C.R., M.M., and P.Z.

## Competing interests

The authors declare no competing interests.
