## [Peer Review File · Nature Communications]

Reviewers' comments:

Reviewer #1 (Remarks to the Author):

Addi et al. present an interesting manuscript examining ESCRT-III-dependent abscission in which they identify new components of the midbody and examine the role of an ALIX-interacting transmembrane protein complex (Syndecan/Syntenin) in positioning ESCRT-III at the abscission site. The data are generally clear and convincing and the experiments performed well. I think the concept is interesting, but have some concerns as outlined below:

1. The concept of membrane anchoring of ESCRT-III to a transmembrane protein is novel, but I wonder if the authors could outline if other membrane proteins were discovered in these preparations? Centralspindlin has been identified previously as being required for tethering the plasma membrane to midbodies (PMID23235882 and related), is there any redundancy or synergy with your system? The concept of an ESCRT-III-membrane anchor is brought up many times, but experimental support for a tethering function was weak in this manuscript. Can the authors show that in the absence of Syntenin/Syndican, the PM is less well tethered to the midbody? Can an artificial tether rescue your phenotype?
2. The molecular analysis of the ALIX-Syntenin-Syndecan recruitment is strong, but to my mind a disconnect exists here: I cannot see how ALIX F676 can be both essential for localising the abscission machinery to the cut site, yet dispensable for abscission itself (PMID18641129). I think the authors should examine the requirement for this site (and for Syntenin/Syndecan) in cytokinesis in their system by quantifying the degree of multinucleation upon knockdown/rescue. Moreover, the phenotype they describe for ALIX depletion is relatively weak, manifesting as a delay in abscission, whereas previous reports in these cell lines describe significant cytokinesis failure. Are you depleting the proteins enough to effect cytokinesis failure?
3. Related to the cytokinesis phenotype, the abscission time reported in control treatments is very long (e.g., Figure 4, >4 hours). Compared to published reports of approx. 70-90 mins (e.g. PMID26929449), it seems that your cells are really struggling to divide without any manipulation. Presentation of representative stills from the timelapse imaging would help understand what is going on here. To distinguish effects on abscission from mitotic progression and midbody formation, could the authors report the time from midbody formation, rather than from furrow ingression? Perhaps a FP-tubulin line would help here, rather than phase imaging.
4. The final set of experiments are a little underwhelming, revealing slight delays to proper GFP-CHMP4B recruitment and abscission. I wonder whether effects are being masked by a parallel recruitment of ESCRT-III via ESCRT-II (PMID26929449). Is CHMP4B recruitment worsened by co-depletion of this alternate pathway? Secondly, can the frequency of recruitment abnormalities be provided (yellow/cyan arrows) and do they persist in a F676D rescue?

Minor

5. The proteomic strategy is novel, elegant and builds well upon the Echard lab's previous work examining the fate of midbody remnants. I think they should avoid describing this as corresponding to midbodies at the time of abscission as the remnants they collect can sit for many hours on the surface and perhaps refer to MB+ and MB- fractions as MBR+ and MBR-.
6. The rescue strategy adopted relies on transient overexpression of CMV-driven RNAi resistant plasmids with a GFP-marker plasmid to detect transfected cells. Can you use antibody staining to report the degree of overexpression of the rescue plasmid and the correlation of GFP-transfection with ALIX/Syntenin/Syndecan transfection?
7. It might be appropriate to include citation of the body of work from Petronczki et al, detailing midbody-plasma membrane tethers.
8. This is just my curiosity, but what are the MKLP- ALIX+ structures (MB-) isolated in your FACS (e.g., Fig 1C)?

Reviewer #2 (Remarks to the Author):

Addi and co-workers have used a novel protocol to purify cytokinetic midbody remnants, and identified 489 enriched proteins by proteomics, of which 150 had already been described to play a role in cytokinesis. The authors chose to focus on three of the most highly enriched proteins, ALIX, syntenin and syndecan-4, which have previously been found to interact in the context of exosome biogenesis. The three proteins were found first at the midbody and then at the abscission zone, and their depletion inhibited abscission. Interactions between the three proteins were found to be important for localization of ESCRT-III to the abscission zone but not to the midbody. The authors conclude that ESCRT-III must be physically connected to an integral membrane protein (syndecan-4) in order to achieve efficient cytokinetic abscission.

This is a timely manuscript that presents a novel and interesting concept for how cytokinetic abscission is mediated, and the data are generally of very good quality. Nevertheless, there are some concerns:

1. ALIX has previously been localized to the midbody and assigned a role in ESCRT-III recruitment (Carlton et al., Science 2008; Morita et al, EMBO J. 2008). The authors now propose that ALIX also has a function at the abscission zone. However, the spinning-disk microscopy imaging does not provide sufficient resolution to conclude with certainty that ALIX is recruited to the abscission zone together with syntenin and syndecan-4. This issue should be confirmed by super-resolution microscopy or electron microscopy.

2. The abscission time phenotypes obtained with syntenin or syndecan-4 knockdown are relatively mild, which argues against the notion that coupling of ESCRT-III to a membrane protein is a “fundamental requirement for efficient scission” as concluded by the authors. Because ALIX and ESCRT-I/-II have been proposed to represent two independent routes for recruitment of ESCRT-III during cytokinetic abscission (Christ et al., JCB 2016), the authors should investigate whether stronger phenotypes could be obtained by simultaneous depletion of TSG101.

3. Although mutational analysis indicates that the ALIX-syntenin-syndecan-4 interactions are required for proper localization of ESCRT-III to the abscission zone, it is not evident that these interactions are actually essential for normal abscission. This important issue needs to be studied in detail.

Reviewer #3 (Remarks to the Author):

In this manuscript, Echard et al. report a proteomic analysis of post-abscission midbodies using a novel protocol to obtain highly purified midbody samples. The midbodies are important organelles where the cleavage of daughter cells takes place during cytokinesis. Many aspects of abscission are not well understood, and this manuscript provides a high quality list of potential candidate proteins that might be involved in the process. The authors follow up on two candidate proteins, syntenin and syndecan-4 and show that they are required to localize the ESCRT-III complex to the abscission site. Overall, the work in this manuscript is well done and provides interesting new insights. It should be published once the following comments have been addressed.

Major comment:

The authors propose that the function of syndecan-4 and syntenin is to tether the ESCRT-III abscission complex to the abscission site via ALIX. This raises the questions of what triggers the association of syntenin/syndecan-4/ALIX during abscission, how they are enriched at this site and what might constrain these tethers, especially since there appear to be large pools of syntenin/syndecan-4 in interphase and dividing cells that are not co-localized. It is beyond the very substantial amount of work already presented in this manuscript to investigate this question in depth, but it would be helpful for the authors to discuss these issues, and to perhaps do some simple experiments to check if common signalling pathways are involved.

Minor comments:

Since MKLP2 is involved in cytokinesis, it would be helpful to include a small control experiment showing that the timing of cytokinesis is similar to wild type cells, to exclude possible artifacts due to MKLP2 overexpression.

Please comment on why two different sample preparations for proteomics would result in somewhat different data sets.

How were the proteins in Fig 1e categorized? GO annotations?

Fig 3C seems to suggest that ALIX overexpression increases syntenin. Is this generally true or just the figure shown?

The authors' model suggests that Alix interacts simultaneously with ESCRTIII and syntenin. If known, a cartoon of interaction domains might be helpful.

The website is a very nice resource for the community and was a very significant effort to put together. However, it needs a little more work to finalize. For example, some literature reference tabs are empty and the color shading could be explained better

There is a small error on p13 line 5: CHO cells are hamster not mouse. Also, Methanol is abbreviated as MetOH not the usual MeOH.

Reviewer #4 (Remarks to the Author):

This manuscript reports the proteomic analyses of post-abscission midbodies (MBs) obtained following FACS sorting of GFP-MKLP2 containing MBs (MB+). To compare the merits of this approach, they determined the relative abundance of protein from MB+ to extracts from particles of the same size but negative for GFP-MKLP2 (MB-) and to MB remnants obtained by differential centrifugation of EDTA treated HeLa cells expressing GFP-MKLP2. From these analyses they further characterized by microscopy a plasma membrane to ESCRT module composed of the proteoglycan syndecan 4 (SDC4), ALIX (CHMP4B) and syntenin-1 (SDCB1) to determine their roles at the abscission site. This study provides valuable insights into cytokinesis and the role of ALIX-SDCB1 to bridge the ESCRT machinery to the plasma membrane through SDC4. Overall, this contribution extends our understanding of the roles of ALIX, SDCB1 and SDC4 in successful abscission. However, there are several points that merit further clarification before this manuscript can be accepted for publication as outlined below.

1. The authors claim that the use of FACS to enrich MB+ protein extracts provides the “purest possible MB remnants” compared to previously reported methods. While examination of Supplementary Table 1 reveals that MB+ extracts are indeed enriched in MKLP1/2, CRIK, PKC1, PLK1 and CEP55, there are several unexpected proteins in TAB3 (e.g. 14-3-3, HSP7C, HSP90, α -enolase, EF-2, Plectin, Talin 1, T complex protein sub-units) that are significantly abundant and their presence remain unexplained. More comprehensive bioinformatics analyses (e.g. GO term enrichment etc...) are needed to fully appreciate the merits of this approach.

2. The immunoblots presented in Figure 1c suggest that MB+ contain by far much lower contaminants from other organelles (e.g. nuclei: H3, ER: calreticulin, and Golgi: GM130) and significant enrichment of MB proteins (e.g. CRIK, CEP55, ALIX, etc...). However, results presented in Supplementary table 1 do not depict such a contrasting enrichment between MB+ and MBE extracts. For example, similar number of peptides were identified for H3, calreticulin or Golgi resident protein GCP60 in both MB+ and MBE extracts by gel or by the eFASP method. The authors should comment on the differences noted.

3. Label-free quantitative proteomics was used to determine the relative changes in protein abundance between different MB enrichment methods. However, the number of replicate is significantly different between gel-based (n=5-6) and eFASP (n=2-3). The relatively low size of eFASP samples limits the statistical significance of differential abundance for these samples. A fold-change of 1.3 was used to define differential abundance, but the ratio appears somewhat arbitrary chosen and does not account for variable FDR expected from the different comparisons. More appropriate statistical analyses are needed.

4. The authors selected proteins ALIX, SDCB1 and SDC4 for functional follow-up experiments. It is unclear how these proteins were selected based on proteomic results since the corresponding #peptides identified is significantly low. The proteomic data appear to be unrelated with the main story of the manuscript detailing the role of these 3 proteins in the ESCRT module. The proteomics section should be rewritten to provide a more linear description that dovetails with the follow-up experiments.

Response to Reviewers

" The Flemmingsome reveals an ESCRT-to-membrane coupling via ALIX / syntenin / syndecan-4 required for completion of cytokinesis "

We thank the four Reviewers for their positive evaluation of our work and for the suggested experiments.

We have carried out the requested experiments and provided new Figures in the revised manuscript to address all their comments.

We believe that the added experimental data and Reviewers' suggestions helped us to strongly reinforce the conclusions of the manuscript.

New experimental data have been added as Fig. 1g, 2c, 3c, 4a, 4b, 4d, 4e, 4f, 4g, 4h, 5e, 5j, Supplementary Fig. 1a, 1e, 4b, 5a, 5b, 5c, 5d, 5h and Supplementary Movie 1.

We provide below a full response to each comment raised by the Reviewers (new experimental figures and text modifications are indicated in blue). We also highlighted changes in the text using track-mode in the revised manuscript.

Reviewer #1 (Remarks to the Author):

Addi et al. present an interesting manuscript examining ESCRT-III-dependent abscission in which they identify new components of the midbody and examine the role of an ALIX-interacting transmembrane protein complex (Syndecan/Syntenin) in positioning ESCRT-III at the abscission site. The data are generally clear and convincing and the experiments performed well. I think the concept is interesting, but have some concerns as outlined below:

1. The concept of membrane anchoring of ESCRT-III to a transmembrane protein is novel, but I wonder if the authors could outline if other membrane proteins were discovered in these preparations?

In the previous version of the manuscript, we indicated that 29 transmembrane proteins were identified as being enriched in the midbody remnant proteome (labeled in blue in Table 1, TAB2). We now further discussed this point on p.14: "This includes (...) transmembrane proteins such as chloride channels (CLIC1 and CLIC4, previously localized at the midbody⁵²), adhesion/tethering/signaling molecules (ITGA3, PlexinB2, ICAM1, BST2, CD44) and tetraspanins (CD9, CD9-P1, TM4SF1) whose potential functions during cytokinesis will be studied in the future."

Centralspindlin has been identified previously as being required for tethering the plasma membrane to midbodies (PMID23235882 and related), is there any redundancy or synergy with your system? The concept of an ESCRT-III-membrane anchor is brought up many times, but experimental support for a tethering function was weak in this manuscript. Can the authors show that in the absence of Syntenin/Syndican, the PM is less well tethered to the midbody? Can an artificial tether rescue your phenotype?

We are sorry if we were not clear enough, but it was not our intention to claim that ALIX/syntenin/syndecan would "tether" the plasma membrane to the midbodies, as indeed previously shown for MgcRacGAP (ref. 57). Actually, we used the word "coupling" to describe that ALIX/syntenin/syndecan allowed the ESCRT-III machinery to closely interact with the plasma membrane at the abscission site, so that ESCRT constriction could promote abscission. New super-resolution microscopy data revealed that both syntenin and ALIX colocalized with CHMP4B at the outer rim of CHMP4B staining (Fig. 2c), a relative localization consistent with the molecular scheme presented in Fig. 2a and with the coupling model in Fig. 6b.

We experimentally ruled out that ALIX/syntenin/syndecan would act as a tether between the plasma membrane and the midbodies, since we observed only a modest increase of binucleated cells (Fig. S5a, see also point #2 below). Indeed, a large amount of binucleated cells would have indicated a role of ALIX/syntenin/syndecan in intercellular bridge stability or midbody integrity, as shown for MgcRacGAP. Time-lapse microscopy further confirmed lack of bridge reopening or midbody integrity defects upon either ALIX, syntenin or syndecan-4 depletion (movies S6-8). Finally, no change in the plasma membrane at the midbody was detected by IF upon depletion of ALIX/syntenin/syndecan (pictures below, with no change in cell mask that strongly labels the midbody, see also Fig. 1d).

We now clearly discuss the difference between PM/midbody tethering and ESCRT/PM coupling in the discussion, on p.16 and specifically PMID23235882 (ref. 56). The central phenotype that we report here is a delay in abscission due to delay in ESCRT-III localization at the abscission site (Fig. 5), due to its unstable localization (Fig. 6) upon ALIX/syntenin/syndecan-4 deletion. The rationale for the model of SDC4 connecting/coupling the ESCRT to the PM comes from 1/the physical interactions (Fig. 2), 2/the requirement of direct interactions (Fig. 3) and 3/the fact that a SDC4 lacking the extracellular domain ECD but retaining the transmembrane domain + the cytoplasmic tail rescues the ESCRT-III localization and abscission delay phenotypes (Fig. 4). To further reinforce the idea of coupling to the PM, we now provide evidence that the cytoplasmic tail of SDC4 alone cannot rescue the phenotype (Fig. S5h). This is consistent with our proposed model: ESCRT-III coupling with the PM depends both on the cytoplasmic tail of SDC4 but also on its TM domain.

We tried to test whether the coupling via syndecan/syntenin might be functionally replaced by an alternative coupling molecule (HIV p6-Gag), but this protein did not localize to the midbody by itself. We have not pursued further since the Editor indicated that "we would not necessarily require new experiments to assess the role of membrane tethering".

2. The molecular analysis of the ALIX-Syntenin-Syndecan recruitment is strong, but to my mind a disconnect exists here: I cannot see how ALIX F676 can be both essential for localising the abscission machinery to the cut site, yet dispensable for abscission itself (PMID18641129). I think the authors should examine the requirement for this site (and for Syntenin/Syndecan) in cytokinesis in their system by quantifying the degree of multinucleation upon knockdown/rescue. Moreover, the phenotype they describe for ALIX depletion is relatively weak, manifesting as a delay in abscission, whereas previous reports in these cell lines describe significant cytokinesis failure. Are you depleting the proteins enough to effect cytokinesis failure?

We experimentally addressed in detail this important point, that was missing in the initial manuscript.

First, we measured the % of binucleated cells upon ALIX, syntenin and syndecan-4 depletion and found a modest, albeit reproducible, increase (Fig. S5a), as mentioned above. For reasons that need further investigation other studies reported a stronger increase in binucleated cells after ALIX depletion, that indeed were rescued by the ALIX F676D mutant (ref. 21-22). As now discussed on p.16, we suspect that this depends on differences in cell lines (e.g. HeLa ATCC cl-2 vs. HeLa Kyoto). In our hands, depleting ALIX in HeLa Kyoto led to a stronger increase in the % of binucleated cells (1.8% in controls to 12.6% in ALIX-depleted cells, $p < 0.0001$). We do not believe that the issue is the level of depletion, since proteins are well depleted in westernblots and IF (see Fig. 3a, 4d, S5c and provided IF in minor points), but rather that proteins involved in the abscission step can lead or not to binucleated cells depending on the cells. Indeed, CEP55 depletion in HeLa ATCC cl-2 cells (used in this study) led to a dramatic increase of cells arrested in cytokinesis (> 50% cells connected by a tubulin-positive bridge and an absence of CHMP4B at the midbody, as expected), yet only 8% of binucleated cells (vs. 2% in controls) (N=3). Our results indicate a role of ALIX in the abscission step, rather than for the stability of the intercellular bridge in HeLa ATCC. We noticed that, except in the brain, ALIX knock-out mice are viable and of correct size, suggesting that ALIX is not essential for cytokinesis although abscission is likely delayed (PMID 28322231). Furthermore, the relative increase in abscission timing upon ALIX depletion reported in Christ et al. seems consistent with the relative increase seen in Fig. 4a ($t_{1/2}$ siCtrl = 70 min, $t_{1/2}$ siALIX = 160 min (thus 2.3-fold increase) compared to a 1.9-fold increase in our cells).

It is unknown why depending on conditions/studies, ALIX depletion leads to variable proportion of binucleated cells vs. cells delayed at the abscission step. However, bridge stability might not need syntenin interaction (as demonstrated by the ALIX F676D mutant in PMID 18641129) yet, syntenin might be important for abscission. We thus now measured the timing of abscission after ALIX depletion and reintroduction of either wild type ALIX or ALIX F676D (Fig. 4a and S5c). In contrast to wild type ALIX, we now report that ALIX F676D was unable to rescue the abscission defects. This indicates that the direct interaction between ALIX and syntenin is required for normal abscission.

As a second approach, we investigated the timing of abscission after syntenin depletion and reintroduction of syntenin Δ ALIX (unable to interact with ALIX). Again, syntenin Δ ALIX could not rescue the abscission defects in contrast to wild type syntenin (Fig. 4b and S5d). For completeness, we also now show that syntenin Δ syndecan (unable to interact with syndecan) could not rescue the abscission defects either (Fig. 4b and S5d).

Altogether, these new results indicate that the direct interaction between ALIX/syntenin on the one hand, and syntenin/syndecan-4 on the other hand are critical for the correct timing of abscission. This is consistent with our previous results showing that direct interactions between ALIX/syntenin/syndecan-4 are required for ESCRT-III localization at the abscission site, using these mutants (Fig. 5a-d).

3. Related to the cytokinesis phenotype, the abscission time reported in control treatments is very long (e.g., Figure 4, >4 hours). Compared to published reports of approx. 70-90 mins (e.g. PMID26929449), it seems that your cells are really struggling to divide without any manipulation. Presentation of representative stills from the timelapse imaging would help understand what is going on here. To distinguish effects on abscission from mitotic progression and midbody formation, could the authors report the time from midbody formation, rather than from furrow ingression? Perhaps a FP-tubulin line would help here, rather than phase imaging.

As requested, we now provide still images to show how we measure abscission timing from a representative video (Fig. S5b). The movie can be found following this link

<http://dl.pasteur.fr/fop/ZabrHlJc/Video%20phase%20contrast.avi>

We don't see a delay in furrow ingression (which takes approximately 20 min) nor in the apparition of the dark midbody structure (which follows furrow ingression) upon ALIX/syntenin/syndecan-4 depletion. Thus, it is the abscission step that is delayed in these conditions.

Over the past 15 years (see PMID 16950109, 21706022, 23948252, 28230050) we obtained consistent results for abscission timing in HeLa ATCC cl-2 cells (mean abscission time 250 min in control siRNA-treated cells). Importantly, we also notice that similar abscission times have been found in other laboratories, including the Prekeris Lab (PMID 24275865, 210 min), Trimble Lab (PMID 21059847, 230 min) or DiCunto Lab (PMID 21849473, 200 min).

Contrary to PMID 26929449, all our measurements are based on actual, physical abscission seen by ICB cut and midbody retraction using phase contrast (see above), rather than FP-tubulin. We believe that this is more accurate, since tubulin staining can be very dim or even absent in cells that take long to divide (PMID 23539606). Thus, it is plausible that the difference of the mean abscission time is essentially due to an underestimation of the cells that take long to divide.

4. The final set of experiments are a little underwhelming, revealing slight delays to proper GFP-CHMP4B recruitment and abscission. I wonder whether effects are being masked by a parallel recruitment of ESCRT-III via ESCRT-II (PMID26929449). Is CHMP4B recruitment worsened by co-depletion of this alternate pathway?

We thank the reviewer for raising the possible contribution of TSG101 (PMID 26929449) in parallel to ALIX in CHMP4B recruitment at the abscission site. We have now investigated this issue in detail and found that it is indeed the case.

First, we confirmed that co-depletion of TSG101 and ALIX led to a very strong delay in abscission (Fig. 4d, e, h), as expected (PMID 26929449). Interestingly, co-depleting either TSG101 and syntenin, or TSG101 and syndecan-4 also strongly delayed abscission (Fig. 4d, f,

g, h). These results indicate that syndecan-4 and syntenin functionally cooperate with TSG101 for abscission. They also suggest that ALIX/syntenin/syndecan-4 and TSG101 act in parallel to promote abscission.

Second, we specifically analyzed CHMP4B recruitment in fixed cells in these conditions. Confirming previous reports (PMID 26929449), we observed that the presence of both TSG101 and ALIX is required for the recruitment of CHMP4B at the midbody itself (Fig. 5e). Strikingly, co-depletion of TSG101 and either syntenin or syndecan-4 did not abolish ESCRT-III recruitment at the midbody (since ALIX is still present) but essentially abolished ESCRT-III localization at the abscission site (Fig. 5e), consistent with the observed aggravated abscission defects (Fig. 4d-h).

Altogether, these new results suggest that ALIX and TSG101 actually cooperate to localize ESCRT-III at the abscission site (our study), and not only to recruit ESCRT-III at the midbody (PMID 26929449). Future studies will be required to understand how ALIX (via syntenin/syndecan-4) and TSG101 (via unknown mechanisms) promote abscission through parallel pathways. These results are presented in Fig. 5d-h, Fig. 5e, described on p.11 and discussed on p.18.

Secondly, can the frequency of recruitment abnormalities be provided (yellow/cyan arrows) and do they persist in a F676D rescue?

We now provide the frequency of abnormalities upon ALIX/syntenin/syndecan-4 depletion showing that there are indeed defects in CHMP4B behavior in these conditions (Fig. 5j). Note that for the new rescue experiments presented in Fig. 4a, 4b, S5c and S5d, we established transduced-cell lines using GFP-tagged constructs, and for time constraints we have not transduced ALIX F576D-mcherry in the HeLa CHMP4B-GFP cell line used in Fig. 5. Since ALIX F676D essentially abolishes syntenin localization at the bridge (as much as upon syntenin depletion) (Fig. 3c, d), we expect the same defects for CHMP4B behavior.

Minor

5. The proteomic strategy is novel, elegant and builds well upon the Echard lab's previous work examining the fate of midbody remnants. I think they should avoid describing this as corresponding to midbodies at the time of abscission as the remnants they collect can sit for many hours on the surface and perhaps refer to MB+ and MB- fractions as MBR+ and MBR-.

We agree with the Reviewer since perhaps some proteins are degraded between abscission and MBR collection, we now wrote that "MBRs correspond to post-abscission midbodies". As suggested, we also changed MB+, MB- and MBE to MBR+, MBR- and MBRE, respectively, throughout the text and Figures.

6. The rescue strategy adopted relies on transient overexpression of CMV-driven RNAi resistant plasmids with a GFP-marker plasmid to detect transfected cells. Can you use antibody staining to report the degree of overexpression of the rescue plasmid and the correlation of GFP-transfection with ALIX/Syntenin/Syndecan transfection?

We show below representative images of ALIX/syntenin/syndecan-4 staining after co-transfection with GFP (> 90% of green cells also express the transgene). We can include this data if the Reviewer feels that it is necessary (they would be available online with the Rebuttal).

7. It might be appropriate to include citation of the body of work from Petronczki et al, detailing midbody-plasma membrane tethers.

As mentioned above (point #1), we have now discussed this paper on p.16.

8. This is just my curiosity, but what are the MKLP- ALIX+ structures (MB-) isolated in your FACS (e.g., Fig 1C)?

We want to investigate this in detail in the future. These likely correspond to large extracellular vesicles, based on our preliminary SEM pictures. We also occasionally found MBRs negative for MKLP2 in this fraction (as expected, since a few percent of the MKLP2-GFP cell line has very weak or absent GFP signal).

Reviewer #2 (Remarks to the Author):

Addi and co-workers have used a novel protocol to purify cytokinetic midbody remnants, and identified 489 enriched proteins by proteomics, of which 150 had already been described to play a role in cytokinesis. The authors chose to focus on three of the most highly enriched proteins, ALIX, syntenin and syndecan-4, which have previously been found to interact in the context of exosome biogenesis. The three proteins were found first at the midbody and then at the abscission zone, and their depletion inhibited abscission. Interactions between the three proteins were found to be important for localization of ESCRT-III to the abscission zone but not to the midbody. The authors conclude that ESCRT-III must be physically connected to an integral membrane protein (syndecan-4) in order to achieve efficient cytokinetic abscission.

This is a timely manuscript that presents a novel and interesting concept for how cytokinetic abscission is mediated, and the data are generally of very good quality. Nevertheless, there are some concerns:

1. ALIX has previously been localized to the midbody and assigned a role in ESCRT-III recruitment (Carlton et al., Science 2008; Morita et al, EMBO J. 2008). The authors now propose that ALIX also has a function at the abscission zone. However, the spinning-disk microscopy imaging does not provide sufficient resolution to conclude with certainty that ALIX is recruited to the abscission zone together with syntenin and syndecan-4. This issue should be confirmed by super-resolution microscopy or electron microscopy.

We indeed report a second pool of endogenous ALIX that was not explicitly described before in PMID 17853893, 18641129 and 18948538. The cone-like structure extending from the midbody to the midbody side (see Fig. 2b) points toward a constricted and often interrupted staining of tubulin, approximately 1-2 μm from the midbody and that has previously been described as the future abscission site or "secondary ingression site" (Ref. 12, 25, 39 and PMID 29588396). Given the resolution of our confocal microscope (roughly $\lambda/2$, thus 200 nm in green), we are confident that we can distinguish the midbody pool from this secondary pool. Actually, CEP55 (which has been reported to localize only at the midbody) was never detected at the abscission site (see Fig. S5f, arrowheads).

To further establish that ALIX indeed localizes to the future abscission site, we now provide time-lapse spinning-disk confocal movie of ALIX-LAP-GFP and SiR-Tubulin Cy5 (Fig. S4b and movie 1). These movies demonstrate that ALIX forms a cone-like structure extending on one side of the midbody (secondary pool) and that abscission occurs at the tip of this cone at a distance of approximately 2 μm from the midbody (arrow in Fig. S4b). The ALIX-mScarlet + CHMP4B-GFP movies (Movie S2) also confirmed that both proteins co-localized to the tip of the cone structure, where abscission took place. Altogether, we conclude that ALIX is localized at the future abscission site, until abscission.

We nevertheless took the opportunity to look at ALIX/CHMP4B/tubulin and syntenin/CHMP4B/tubulin using super-resolution (SIM) to better describe the spatial relationship between these three proteins in the pool pointing to the abscission site. Remarkably, both syntenin and ALIX colocalized with CHMP4B at the outer rim of CHMP4B staining (Fig. 2c). This relative localization is fully consistent with the molecular scheme presented in Fig. 2a and with the coupling model in Fig. 6b.

2. The abscission time phenotypes obtained with syntenin or syndecan-4 knockdown are relatively mild, which argues against the notion that coupling of ESCRT-III to a membrane protein is a "fundamental requirement for efficient scission" as concluded by the authors. Because ALIX and ESCRT-I/-II have been proposed to represent two independent routes for recruitment of ESCRT-III during cytokinetic abscission (Christ et al., JCB 2016), the authors should investigate whether stronger phenotypes could be obtained by simultaneous depletion of TSG101.

We thank the reviewer for raising the possible contribution of TSG101 (PMID 26929449) in parallel to ALIX in CHMP4B recruitment at the abscission site. We have now investigated in detail this issue and found that it is indeed the case.

First, we confirmed that co-depletion of TSG101 and ALIX led to a very strong delay in abscission (Fig. 4d, e, h), as expected (PMID 26929449). Interestingly, co-depleting either TSG101 and syntenin, or TSG101 and syndecan-4 also strongly delayed abscission (Fig. 4d, f, g, h). These results indicate that syndecan-4 and syntenin functionally cooperate with TSG101 for abscission. They also suggest that ALIX/syntenin/syndecan-4 and TSG101 act in parallel to promote abscission.

Second, we specifically analyzed CHMP4B recruitment in fixed cells in these conditions. Confirming previous reports (PMID 26929449), we observed that the presence of both TSG101 and ALIX is required for the recruitment of CHMP4B at the midbody itself. Strikingly, co-depletion of TSG101 and either syntenin or syndecan-4 did not abolish ESCRT-III recruitment at the midbody (since ALIX is still present) but essentially abolished ESCRT-III localization at the abscission site (Fig. 5e), consistent with the observed aggravated abscission defects (Fig. 4d-h).

Altogether, these new results suggest that ALIX and TSG101 actually cooperate to localize ESCRT-III at the abscission site (our study), and not only to recruit ESCRT-III at the midbody (PMID 26929449). Future studies will be required to understand how ALIX (via syntenin/syndecan-4) and TSG101 (via unknown mechanisms) promote abscission through parallel pathways. These results are presented in Fig. 5d-h, Fig. 5e, described on p.11 and discussed on p.18.

We changed the term "fundamental" to "critical" in the initial sentence. It now reads: "Thus, our study suggests that the coupling of the ESCRT-III machinery to a membrane protein via ALIX-syntenin or equivalent modules represents a critical requirement for efficient scission during cytokinesis, exosome formation and retroviral budding." (p.18)

3. Although mutational analysis indicates that the ALIX-syntenin-syndecan-4 interactions are required for proper localization of ESCRT-III to the abscission zone, it is not evident that these interactions are actually essential for normal abscission. This important issue needs to be studied in detail.

As requested, we have now directly quantified abscission timing in cells expressing wild type vs. point mutants to disrupt the interactions between ALIX/syntenin and syntenin/syndecan.

First, we found that ALIX F676D was unable to rescue the abscission defects, in contrast to wild type ALIX (Fig. 4a and S5c). This indicates that the direct interaction between ALIX and syntenin is required for normal abscission.

Second, we investigated the timing of abscission after syntenin depletion and reintroduction of syntenin Δ ALIX (unable to interact with ALIX). Again, syntenin Δ ALIX could not rescue the defects, in contrast to wild type syntenin (Fig. 4b and S5d).

Third, we also now show that syntenin Δ syndecan (unable to interact with syndecan) could not rescue the abscission defects either (Fig. 4b and S5d).

Altogether, these new results indicate that the direct interaction between ALIX/syntenin on the one hand, and syntenin/syndecan-4 on the other hand are critical for the correct timing

of abscission. This is consistent with our previous results showing that direct interactions between ALIX/syntenin/syndecan-4 are required for ESCRT-III localization at the abscission site, using these mutants (Fig. 5a-d).

[Please note that we changed MB+, MB- and MBE to MBR+, MBR- and MBRE, respectively, throughout the text and Figures, as suggested by Reviewer #1]

Reviewer #3 (Remarks to the Author):

In this manuscript, Echard et al. report a proteomic analysis of post-abscission midbodies using a novel protocol to obtain highly purified midbody samples. The midbodies are important organelles where the cleavage of daughter cells takes place during cytokinesis. Many aspects of abscission are not well understood, and this manuscript provides a high quality list of potential candidate proteins that might be involved in the process. The authors follow up on two candidate proteins, syntenin and syndecan-4 and show that they are required to localize the ESCRT-III complex to the abscission site. Overall, the work in this manuscript is well done and provides interesting new insights. It should be published once the following comments have been addressed.

Major comment:

The authors propose that the function of syndecan-4 and syntenin is to tether the ESCRT-III abscission complex to the abscission site via ALIX. This raises the questions of what triggers the association of syntenin/syndecan-4/ALIX during abscission, how they are enriched at this site and what might constrain these tethers, especially since there appear to be large pools of syntenin/syndecan-4 in interphase and dividing cells that are not co-localized. It is beyond the very substantial amount of work already presented in this manuscript to investigate this question in depth, but it would be helpful for the authors to discuss these issues, and to perhaps do some simple experiments to check if common signalling pathways are involved.

We have now discussed this interesting question on p.16. Please note that the Editor indicated that "we would not necessarily require new experiments to assess the role of other signalling pathways". ALIX/syntenin/syndecan-4 all localize concomitantly at the midbody and at the abscission site (Fig. 2 and Movies S2-4). Yet, as noticed by the Reviewer, we did not detect any ALIX or syntenin at the plasma membrane in the cell body, although syndecan-4 is clearly localized there (Movie S4). This suggests that post-translational modifications of either ALIX, syntenin and/or syndecan-4, perhaps via phosphorylations (Ref. 43, 57) regulate the formation of the tripartite complex at the midbody and at the abscission site during cytokinesis (p.16). Syndecan-4 also binds to PtdIns(4,5)P2 and a local increase of this lipid at the midbody might also regulate the formation of the complex (PMID 12377772).

Minor comments:

Since MKLP2 is involved in cytokinesis, it would be helpful to include a small control experiment showing that the timing of cytokinesis is similar to wild type cells, to exclude possible artifacts due to MKLP2 overexpression.

As requested, we measured the timing of abscission in the MKLP2-GFP HeLa cell line that we used for MBR purification. As shown in Fig. S1a, there is no difference in abscission, as compared to the control HeLa cells.

Please comment on why two different sample preparations for proteomics would result in somewhat different data sets.

The midbody is constituted of highly dense and structured proteins that required 2% SDS for proper protein denaturation and efficient solubilization (PMID 7130277). SDS removal is an essential prerequisite for achieving peptide digestions required for the mass-spectrometric analysis. It can be accomplished by numerous techniques such as protein precipitation, strong cation exchange, protein and peptide level purification with pierce detergent removal cartridges and eFASP etc.

In this work, two approaches were used to increase the high-throughput of our proteomics analysis: eFASP (enhanced Filter-Aided Sample Preparation) and in-gel digestion (see ref. 48 for a detailed comparison). As compared to in-gel digestion, eFASP enables better tryptic digestion efficiency thanks to the substitution of 0.2% deoxycholic acid surfactant for urea during the digestion step and generally allows to obtain a better proteome coverage for both cytosolic and membrane proteins (ref. 48). Nevertheless, some proteins are only found with the in-gel approach and, as illustrated in Fig. S1d, both methods are complementary (indicated on p.6).

How were the proteins in Fig 1e categorized? GO annotations?

We suppose that the Reviewer meant Fig. 1f. For Fig. 1f, we used the STRING Functional Protein Association database to reveal protein-protein interactions in the *Enriched Flemmingsome*. Two proteins are linked if phylogenetic co-occurrence interactions, experimentally determined interactions or database annotated interactions have been reported by STRING. Confidence in interactions are represented by the widths of the edges, which is a function of the STRING combined score (score computed from all these kinds of interactions). We colored each category ("CPC", "ESCRT", etc) based on manually-curated published literature if the protein is part of the CPC or ESCRT complex. This has been now explained in the Fig 1f legend.

We now provide in Fig. 1g a categorization of all proteins of the *Total Flemmingsome* (MB+) using 11 clusters of GO terms (see Table 1 TAB7, for the GO terms included in each cluster), such as "Mitosis/Cell division", "Actin-related", "Traffic/Transport/Vesicle" etc. Of note, these GO-clusters covered the vast majority (87.5%) of the proteins identified in MBR+.

As seen in Fig. 1g, the category "Mitosis/Cell division" is significantly enriched in MB+ as compared to the total identified proteins. Importantly, this is also true for the proteins more abundant in MB+ as compared to MBE (Fig. S1e). Altogether, this GO term analysis indicates that flow cytometry-based MB+ purification is superior to MBE purification for obtaining proteins known to be implicated in cell division, as expected. This GO analysis is now discussed on p.7 of the manuscript.

Fig 3C seems to suggest that ALIX overexpression increases syntenin. Is this generally true or just the figure shown?

Syntenin intensity can sometimes be variable and we did not see a general increase of syntenin upon ALIX overexpression. We thus replaced the picture.

The authors' model suggests that Alix interacts simultaneously with ESCRTIII and syntenin. If known, a cartoon of interaction domains might be helpful.

The Zimmermann Lab previously showed that ALIX/syntenin and syntenin/syndecan-4 directly interacted *in vitro* in the context of exosome formation (Ref. 41). In the initial version of the manuscript, we had summarized in Fig. 2a the exact domains of interactions (as well as the mutations that can disrupt these interactions) with a cartoon.

The website is a very nice resource for the community and was a very significant effort to put together. However, it needs a little more work to finalize. For example, some literature reference tabs are empty and the color shading could be explained better.

We agree that the color code was not obvious and we changed it. In the new version of the webpage, we now explain better the color code and also the details of the different elements of the webpage, in the "Help" Tab on the right side of the webpage.

There are two buttons (on the left) for choosing either the list of proteins of the *Total Flemmingsome* (white background) or the list of proteins of the *Enriched Flemmingsome* (green background).

In the *Enriched Flemmingsome*, we indicated for each protein whether there are published references reporting if the protein has been localized to the midbody during cytokinesis and/or has been functionally involved in cytokinesis in mammalian cells. The proteins with published references are written in magenta (with green background), the proteins with no published references related to midbody or cytokinesis are left in black (with green background). The non-enriched proteins from the *Total Flemmingsome* were not browsed individually (that is why no references were added to them) but they are linked to the Uniprot website.

There is a small error on p13 line 5: CHO cells are hamster not mouse. Also, Methanol is abbreviated as MetOH not the usual MeOH.

Thanks for pointing out these mistakes. They have been corrected.

[Please note that we changed MB+, MB- and MBE to MBR+, MBR- and MBRE, respectively, throughout the text and Figures, as suggested by Reviewer #1]

Reviewer #4 (Remarks to the Author):

This manuscript reports the proteomic analyses of post-abscission midbodies (MBs) obtained following FACS sorting of GFP-MKLP2 containing MBs (MB+). To compare the merits of this approach, they determined the relative abundance of protein from MB+ to extracts from particles of the same size but negative for GFP-MKLP2 (MB-) and to MB remnants obtained by differential centrifugation of EDTA treated HeLa cells expressing GFP-MKLP2. From these analyses they further characterized by microscopy a plasma membrane to ESCRT module composed of the proteoglycan syndecan 4 (SDC4), ALIX (CHMP4B) and synthenin-1 (SDCB1) to determine their roles at the abscission site. This study provides valuable insights into cytokinesis and the role of ALIX-SDCB1 to bridge the ESCRT machinery to the plasma membrane through SDC4. Overall, this contribution extends our understanding of the roles of ALIX, SDCB1 and SDC4 in successful abscission. However, there are several points that merit further clarification before this manuscript can be accepted for publication as outlined below.

1. The authors claim that the use of FACS to enrich MB+ protein extracts provides the “purest possible MB remnants” compared to previously reported methods. While examination of Supplementary Table 1 reveals that MB+ extracts are indeed enriched in MKLP1/2 CR1K, PKC1, PLK1 and CEP55, there are several unexpected proteins in TAB3 (e.g. 14-3-3, HSP7C, HSP90, a-enolase, EF-2, Plectin, Talin 1, T complex protein sub-units) that are significantly abundant and their presence remain unexplained. More comprehensive bioinformatics analyses (e.g. GO term enrichment etc...) are needed to fully appreciate the merits of this approach.

New proteins identified in MB+

The TAB3 (Table 1) compiles the totality of the proteins that have been identified by mass spectrometry, whether they are in the Total cell lysate, in the MB- fraction, in the MB+ fraction (= *Total Flemmingsome*) or in the MBE fraction (= MBs only enriched by centrifugation).

Concerning the list of proteins that the Reviewer finds unexpected, all are indeed found in MB+ (*Total Flemmingsome*, Table 1 TAB1) but only 3 are significantly enriched and belong to the *Enriched Flemmingsome* (14-3-3, HSP7C and HSP90, see TAB2, Table 1).

We are not surprised that cytosolic proteins (such as a-enolase, EF-2, Plectin, Talin, T complex protein subunits) can be detected in the *Total Flemmingsome*. Indeed, midbody remnants contain part of the cytosol that was present when cytokinetic abscission occurred. As such, these proteins are present but not enriched, as they were trapped at the time of abscission.

Interestingly, the 3 proteins pointed out by the Reviewer which are found in the *Enriched Flemmingsome* belong to protein families that have all been already localized and/or implicated in cytokinesis.

- 14-3-3 sigma controls the amount of PLK1 at the midbody and is required for cytokinesis (PMID 17361185, PMID18604201). In addition, 14-3-3 eta localizes to the cytokinetic bridge (PMID23547714). These references are indicated in the companion website.

- HSP90 and HSP7C (HSPA8) have not yet been implicated in cytokinesis. However, other heat shock proteins such as HSPB1 play a role in cytokinesis (PMID28166285) and the isoforms HSPA14 and HSPA1B have been localized to the cytokinetic bridge (PMID18841484). Actually,

HSPB1, HSPA14 and HSPA1B have also been successfully identified in the *Enriched Flemmingsome*.

We thus believe that HSP90 and HSP7C (and other identified heat shock proteins) are interesting hits to study in detail in the future. Dr Ahna Skop presented at the ASCB meetings 2018 and 2019 that local translation occurred in midbodies (which is fully consistent with the biosynthesis/translation hits identified in the *Enriched Flemmingsome*, Fig. 1e and 1f), and it is conceivable that chaperones of the heat shock family are locally involved in protein folding.

Bioinformatic analysis

As requested, we now provide a more complete description and quantification of the merit of the flow cytometry-based purification that we developed, as compared to conventional MB enrichment methods by centrifugation (see below). To avoid excessive claims, we also changed "To obtain the purest possible MBRs, we developed an original protocol for isolating fluorescent GFP-positive MBRs ("MBR+")..." to "In order to improve the enrichment of MBRs, ..." (p.5).

First, to quantify whether the MB+ proteome is more enriched than the MBE or Total in proteins relevant for cell division, we first grouped GO terms in 11 clusters (see Table 1 TAB7, for the GO terms included in each cluster). Of note, these GO-clusters ("Mitosis/Cell division", "Actin-related" etc.) covered the vast majority (87.5%) of the proteins identified in MB+ (Fig. 1g). Then, we conducted an analysis of the clusters of GO terms that are over-represented in MB+ as compared to either total identified proteins (Fig. 1g) or MBE (Fig. S1e), using hypergeometric tests. Results are displayed as histograms representing the number of corresponding proteins associated to each cluster of GO terms and the confidence in the enrichment ($[\log_{10}(\text{p-values})]$ represented as a colour gradient depending on the enrichment p-values, as indicated (if the p-value is superior to 10% then the bar is grey and the cluster can be considered not significantly enriched). As seen in Fig. 1g, the category "Mitosis/Cell division" is significantly enriched in MB+ as compared to the total identified proteins. Importantly, this is also true for the proteins more abundant in MB+ as compared to MBE (Fig. S1e). Interestingly, a number of proteins clustered in GO categories related to actin, microtubule and membrane trafficking, which play key roles in cytokinesis (see PMID29689230, PMID29438904), are also more significantly enriched in MB+. Altogether, this GO term analysis indicates that flow cytometry-based MB+ purification is superior to MBE purification for obtaining proteins known to be implicated in cell division.

Second, to further assess this conclusion, we took advantage of the statistical analysis previously presented in Fig. S2 as individual Volcano plots (separately analyzed for the eFASP and Gel MS data). In these plots, the red dots illustrate the MB+ proteins significantly enriched when compared to the three controls. Interestingly, based on the literature, it appears that the 15 proteins with the best MB+/MBE ratio using eFASP extraction were all known to play a role in cytokinesis. They correspond to well-established cytokinetic proteins, for instance PRC1, CEP55, INCENP, KIF23, CitronKinase etc. These proteins were found enriched 6- to 226-fold in MB+ as compared to MBE (Fig. S2, middle panels and Table 1, TAB2). Similarly, among the 15 proteins with the best MB+/MBE ratio using gel extraction, 13 are directly implicated in cytokinesis, 1 has not been implicated and the other is actually syntenin. Here again, these proteins were enriched 3-to 36-fold in MB+ as compared to MBE (Fig. S2, middle panels and Table 1, TAB2). Thus, MB+ proved better than MBE in identifying core, established proteins in cytokinesis.

We now describe and discuss in the text the merit of flow cytometry purification over conventional, centrifugation-based MBR purification (p.7) and we provide the GO analysis in Fig. 1g and Fig. S1e.

2. The immunoblots presented in Figure 1c suggest that MB+ contain by far much lower contaminants from other organelles (e.g. nuclei: H3, ER: calreticulin, and Golgi: GM130) and significant enrichment of MB proteins (e.g. CRIK, CEP55, ALIX, etc...). However, results presented in Supplementary table 1 do not depict such a contrasting enrichment between MB+ and MBE extracts. For example, similar number of peptides were identified for H3, calreticulin or Golgi resident protein GCP60 in both MB+ and MBE extracts by gel or by the eFASP method. The authors should comment on the differences noted.

Regarding the contaminants, we agree there is not a large difference in the number of peptides identified in MBE as compared to MB+. However, even when the same number of peptides is identified for a given protein, these peptides could actually be more abundant in one sample than in the other.

The relative abundance of a protein was thus measured by examining the LFQ intensities and not only the number of identified peptides. When examining the results of the differential analyses based on the LFQ intensities, all the contaminant proteins cited by the Reviewer appeared either less abundant in MB+ than in MBE, or were not even detected in MB+ (Table 1, LFQ values on the right part of the TAB3, columns AK-BP).

Specifically:

-The protein histone H3 was found significantly less abundant in MB+ than in MBE ($\log_2(\text{MB+}/\text{MBE})=-2$) using eFASP, and it could not be quantified in MB+ using Gels:

Differential analysis: MB+ vs MBE (LFQ intensity)			
eFASP		Gel	
$\log_2(\text{MB+}/\text{MBE})$	pvalue	$\log_2(\text{MB+}/\text{MBE})$	pvalue
-2.003263736	0.00135223	n.a.	n.a.

[n.a. means “not available” because the LFQ values for the identified peptides were below the threshold required for quantification (minimum of 2 unique peptides)].

-The protein Calreticulin was also found less abundant in the MB+ than in MBE:

Differential analysis: MB+ vs MBE (LFQ intensity)			
eFASP		Gel	
$\log_2(\text{MB+}/\text{MBE})$	pvalue	$\log_2(\text{MB+}/\text{MBE})$	pvalue
-2.542308244	3.17791E-05	-0.675804841	0.10225605

- Finally, the Golgi protein GCP60 could not be quantified in MB+ (but was only quantified in MB- using Gels and in Total using eFASP):

Differential analysis: MB+ vs MBE (LFQ intensity)			
eFASP		Gel	
log2(MB+/MBE)	pvalue	log2(MB+/MBE)	pvalue
n.a.	n.a.	n.a.	n.a.

Of note, we did not blot for GCP60 in Fig. 1c and corresponding blot S1, but for another Golgi marker (GM130). GM130 was not detected in MB+ or MBE using mass spectrometry, and it was detected in MBE but not in MB+ using immunoblots.

The above eFASP LFQs indicate that Histone H3 and Calreticulin were reduced at least 4-fold in MB+ as compared to MBE.

To compare in MBE vs. MB+ the levels of the different contaminants by immunoblots, we provided serial dilutions for accurate comparisons. As shown in the immunoblots in Fig. S1c (previously S1b), we observed that there was no signal detected for Histone H3 after a 4-fold dilution of MBE fraction. Similarly, almost no signal was detected for Calreticulin after a 4-fold dilution of MBE fraction. Thus, an absence of signal in MB+ in the immunoblot presented in the corresponding Fig. 1c revealed a >4-fold reduction of these intracellular markers, as compared to MBE, but not necessarily a complete absence (the limit of detection having been reached). Altogether, the results of the Table1 are consistent with the immunoblots.

To better reflect that low levels of contaminants may be present, we changed "no significant contamination" to "reduced contamination" when we describe the results of the immunoblots, and the sentence now reads: "Western blot analysis demonstrated that the MB+ population contained highly enriched known midbody proteins [MKLP1, CRIK, PRC1, PLK1, CEP55] and showed reduced contamination, as compared to MB-, total cell lysate (Tot) and MBE fractions, with intracellular compartments [Calreticulin (Endoplasmic Reticulum), GM130 (Golgi), Tom22 (mitochondria), HistoneH3 (nucleus), EEA1 (endosomes)] (Fig. 1c and Supplementary Fig. 1c)." (p.5).

3. Label-free quantitative proteomics was used to determine the relative changes in protein abundance between different MB enrichment methods. However, the number of replicate is significantly different between gel-based (n=5-6) and eFASP (n=2-3). The relatively low size of eFASP samples limits the statistical significance of differential abundance for these samples. A fold-change of 1.3 was used to define differential abundance, but the ratio appears somewhat arbitrary chosen and does not account for variable FDR expected from the different comparisons. More appropriate statistical analyses are needed.

The difference in sample size has been considered for the statistical analysis of each preparation method (eFASP and in-gel analysis) presented in Table 1, as explained below.

We would first like to emphasize that we used two complementary extraction methods (eFASP and gel-based) in order to maximize the number of proteins identified in MB+ (see ref.48). In our study, we did not aim to compare the merit of one extraction method vs. the other; rather,

we compiled all the statistically more abundant proteins in MB+ compared to either Total, MB- or MBE.

For each comparison, we conducted two totally independent differential analyses, one using eFASP samples on the one hand and one using gel samples on the other hand (Fig. S1d, S2). For each comparison, statistical analyses have been conducted both using a fold-change threshold of 1.3 between conditions (MB+ vs MB-, MB+ vs MBE and MB+ vs Tot), and also using a threshold on p-values determined from an adaptive Benjamini-Hochberg procedure to get a false discovery rate (FDR) below 5% among discoveries (see the material and methods section). We account for variable FDR expected from the different comparisons since the adaptive Benjamini-Hochberg procedure has been applied to get adjusted p-values for each comparison.

Note that the values we chose (1.3-fold change and FDR 5%) are always arbitrary and no rule has been imposed in the literature (see PMID31301518 for a discussion of this subject). We decided to set the fold-change threshold to 1.3 since a number of cytokinesis essential proteins (such as the RhoA GTPase, many actin and trafficking proteins, all the ESCRT proteins etc.) are also present in interphase cells (and thus in the Total cell extract). By setting the threshold too high, we would have lost key proteins potentially involved in cytokinesis in the *Enriched Flemmingsome*, but we agree that this is indeed an arbitrary choice (now discussed on p.15). Nevertheless, note that the list of all proteins identified in the MB+ (the *Total Flemmingsome*) is available in Table 1 (TAB1) and can be used by other investigators to test their functional relevance in cytokinesis, whatever their magnitude of enrichment.

4. The authors selected proteins ALIX, SDCB1 and SDC4 for functional follow-up experiments. It is unclear how these proteins were selected based on proteomic results since the corresponding #peptides identified is significantly low. The proteomic data appear to be unrelated with the main story of the manuscript detailing the role of these 3 proteins in the ESCRT module. The proteomics section should be rewritten to provide a more linear description that dovetails with the follow-up experiments.

The aim of our proteomic study was to identify new potential hits for cytokinesis using an original purification method, and to characterize a few of them. Since proteins involved in cytokinesis are enriched in the midbody, we focused on proteins significantly enriched in MB+ as compared to control fractions. Based on MB+/control fold enrichment (calculated from LFQs, see Table 1 TAB3, columns BQ-CN) and statistical significance, we defined the *Enriched Flemmingsome* (Table 1, TAB2), which is summarized in Fig. 1e.

As indicated in Fig. 1e (top panel), ALIX (PDCD6I) and syntenin (SDCBP) are among the most enriched and significant proteins identified in our screen. Dots in red indicate proteins that have been found more abundant in all 3 comparisons (MB+ vs. Total, MB- and MBE). The chart below summarizes the best ratios (fold-increase) obtained for each comparison (see Table 1, TAB2):

	MB+/MB-	MB+/MBE	MB+/Total
ALIX	2.67	3.52	3.3
syntenin	3.55	6.3	10.0

Concerning SDC4: only 16 proteins are quantified in MB+ but not detected in the 3 other fractions (using either gel or eFASP) and SDC4 is one of them (Fig. 1e, bottom panel).

Altogether, the three proteins ALIX, syntenin and syndecan-4 are among the best candidates for a role in cytokinesis (beyond the known, top proteins of the *Enriched Flemmingsome*). In addition, they are known to form a tri-partite complex in interphase in the context of exosome formation. But whether the same complex could be involved in cell division was unknown. Finally, the implication of ALIX in abscission has been established but its exact role remains elusive. This provided rationale for focusing on these 3 proteins.

As requested, we re-wrote part of the proteomic section to better explain why we focused on these proteins. In particular we added: "In the rest of this study, we decided to focus on ALIX (PDCD6IP)-syntenin (SDCBP)-syndecan-4 (SDC4) (Fig. 2a). Indeed, these three proteins were found among the most enriched in MBR+ compared to all other fractions (Fig. 1e, highlighted in red) and are known to form a tri-partite complex in interphase in the context of exosome formation⁴¹. Whether the same complex could be involved in cell division was unknown and could potentially reinforce the idea that exosome formation and cytokinetic abscission share common basic mechanisms. In addition, the implication of ALIX in abscission has been established (e.g. ref.29) but its exact role remains elusive." (p.7).

[Please note that we changed MB+, MB- and MBE to MBR+, MBR- and MBRE, respectively, throughout the text and Figures, as suggested by Reviewer #1]

REVIEWERS' COMMENTS:

Reviewer #1 (Remarks to the Author):

I think the authors have done a good job in revision, that this is a cool paper and is suitable for publication.

Reviewer #2 (Remarks to the Author):

The authors have successfully addressed the points I raised. I am happy to recommend this very interesting manuscript for publication in Nature Communications.

Reviewer #3 (Remarks to the Author):

The authors have done an excellent job revising the manuscript. I am happy with their response to my comments as well as the other reviewers' comments and recommend publication.

Reviewer #4 (Remarks to the Author):

The authors have responded satisfactorily to my comments and I recommend publication of the manuscript.